# Multiple origins of prokaryotic and eukaryotic single-stranded DNA viruses from bacterial and archaeal plasmids

Darius Kazlauskas [1], Arvind Varsani [2,3], Eugene V. Koonin [4] & Mart Krupovic [5]

Single-stranded (ss) DNA viruses are a major component of the earth virome. In particular, the circular, Rep-encoding ssDNA (CRESS-DNA) viruses show high diversity and abundance in various habitats. By combining sequence similarity network and phylogenetic analyses of the replication proteins (Rep) belonging to the HUH endonuclease superfamily, we show that the replication machinery of the CRESS-DNA viruses evolved, on three independent occasions, from the Reps of bacterial rolling circle-replicating plasmids. The CRESS-DNA viruses emerged via recombination between such plasmids and cDNA copies of capsid genes of eukaryotic positive-sense RNA viruses. Similarly, the *rep* genes of prokaryotic DNA viruses appear to have evolved from HUH endonuclease genes of various bacterial and archaeal plasmids. Our findings also suggest that eukaryotic polyomaviruses and papillomaviruses with dsDNA genomes have evolved via parvoviruses from CRESS-DNA viruses. Collectively, our results shed light on the complex evolutionary history of a major class of viruses revealing its polyphyletic origins.

[1] Institute of Biotechnology, Life Sciences Center, Vilnius University, Saulėtekio av. 7, Vilnius 10257, Lithuania. [2] The Biodesign Center for Fundamental and Applied Microbiomics, School of Life Sciences, Center for Evolution and Medicine, Arizona State University, Tempe, AZ 85287, USA. [3] Structural Biology Research Unit, Department of Integrative Biomedical Sciences, University of Cape Town, Rondebosch, 7700 Cape Town, South Africa. [4] National Center for Biotechnology Information, National Library of Medicine. National Institutes of Health, Bethesda, MD 20894, USA. [5] Department of Microbiology, Institut Pasteur, 25 rue du Docteur Roux, Paris 75015, France. Correspondence and requests for materials should be addressed to M.K. (email: krupovic@pasteur.fr)

Viruses with single-stranded (ss)DNA genomes represent a vast, highly diverse supergroup of medically, ecologically, and economically important pathogens infecting hosts from all three domains of cellular life[1,2]. Although for years, ssDNA viruses have been thought to be relatively rare in the biosphere, recent metagenomics studies have increasingly revealed high abundance of these viruses in diverse environments[3–15]. Currently, ssDNA viruses are classified into 13 families, 9 of which include (presumably) eukaryotic viruses, but many uncultivated ssDNA viruses remain unclassified. The majority of ssDNA viruses (9 families) have small circular genomes, which are known or predicted to be replicated by the rolling-circle mechanism. This mechanism of replication is initiated by the virus-encoded Rep protein of the HUH endonuclease superfamily, characterized by the signature HUH motif, in which two histidine residues are separated by a bulky hydrophobic residue[2,16–19]. Informally, these viruses are often collectively referred to as circular, Rep-encoding ssDNA (CRESS-DNA) viruses[1,17]. A variation on this theme is employed by members of the *Parvoviridae* family which have linear ssDNA genomes replicated by the rolling-hairpin mechanism initiated by a Rep protein homologous to those of the CRESS-DNA viruses[18,20]. Members of the family *Bidnaviridae* apparently have evolved from parvoviruses by replacing the HUH endonuclease domain with the DNA polymerase from polintoviruses[21].

The Rep proteins of ssDNA viruses of prokaryotes (bacteria and archaea) and eukaryotes display distinct domain organizations[2]. In eukaryotic CRESS-DNA viruses, the endonuclease domain is fused to a superfamily 3 helicase (S3H) domain[22], which is responsible for unwinding of the double-stranded (ds)DNA replicative intermediate and, in some viruses, packaging of the viral genome into assembled empty capsids[20,23]. By contrast, none of the bacterial or archaeal ssDNA viruses isolated to date encodes a Rep fused to a helicase domain[2]. Instead, these viruses recruit a cellular helicase for the same function[24]. Such dichotomy in the domain organization of the Rep proteins raises questions regarding the evolutionary relationship between ssDNA viruses infecting hosts from different cellular domains[25,26]. Furthermore, HUH Reps are not restricted to ssDNA viruses, but are also functional in several groups of bacterial and archaeal dsDNA viruses, including certain members of the families *Sphaerolipoviridae*, *Rudiviridae*, *Corticoviridae,* and *Myoviridae*. However, the implications of the presence of this gene for potential evolutionary links between these dsDNA viruses and ssDNA viruses remain unclear.

The HUH endonucleases are also encoded by diverse bacterial and archaeal as well as several eukaryotic plasmids and transposons, some of which have been shown experimentally to replicate and/or transpose via the rolling-circle mechanism[27–30]. The homology between the endonuclease domains of the viral and bacterial plasmid Reps has been initially inferred from the conservation of 3 signature motifs[18,19], and subsequently validated by structural analyses[16]. Motif I, UUTU (U denotes hydrophobic residues), is thought to be involved in the recognition of the origin of replication. Motif II, HUH, is involved in the coordination of divalent metal ions, $Mg^{2+}$ or $Mn^{2+}$, which are essential for endonuclease activity at the origin of replication[18,31]. Motif III (YxxK/YxxKY, where x is any amino acid) is involved in dsDNA cleavage and subsequent covalent attachment of the Rep through the catalytic tyrosine residue to the 5' end of the cleaved product[16,18,32]. The HUH endonucleases encoded by prokaryotic plasmids, viruses and transposons can have either two or one catalytic tyrosine residue in the motif III, whereas all known eukaryotic Rep-encoding viruses contain a single tyrosine residue[18]. Notably, whereas most prokaryotic Reps consist of stand-alone endonuclease domains, some bacterial plasmids encode Reps with the domain organization similar to that characteristic of eukaryotic ssDNA viruses, that is, a nuclease-helicase fusion. For example, it has been shown that Reps encoded by plasmids of phytoplasma, plant-pathogenic bacteria, show the highest sequence similarity to Reps of plant-infecting geminiviruses[33,34]. However, whether the similar domain organization is a result of convergent evolution or whether it alludes to a more recent common ancestry of the corresponding replicons remained unclear.

Here, we systematically explore the relationships among Rep-encoding DNA viruses, plasmids, and transposons from all three cellular domains. We identify 8 previously undescribed families of integrative plasmids that are widespread across different bacterial phyla and show that they have seeded the eukaryotic CRESS-DNA virosphere on at least 3 independent occasions. Similarly, the origins of bacterial and archaeal ssDNA viruses replicating by the rolling-circle mechanism can be traced to different families of prokaryotic plasmids, emphasizing tight evolutionary connections between viruses and capsid-less mobile genetic elements (MGE).

## Results

**Global network of the HUH replicons**. To explore the evolutionary history of the HUH replicons, we collected a dataset of HUH endonucleases—the only protein encoded by all these replicons—representing each family of viruses, plasmids, and transposons associated with hosts across all three cellular domains[16,27–30]. In this analysis, we did not consider Mob relaxases involved in plasmid conjugation. Enzymes in this family encompass circularly permuted conserved motifs which complicate their sequence-based comparison with the HUH endonucleases involved in DNA replication or transposition[16,19]. The resulting dataset included 8764 sequences. These were grouped based on pairwise similarity, and clusters were identified using a convex clustering algorithm (p-value threshold of 1e−08) with CLANS[35]. This analysis revealed 33 clusters which varied in size from 7 to 2711 sequences (Supplementary data 1). Following an inspection of the connectivity between clusters (Fig. 1), we defined 2 orphan clusters and 2 superclusters, which displayed either no or very few connections to each other (Supplementary data 1). Nevertheless, comparison of the available high-resolution structures for representatives of both orphan clusters and the 2 superclusters[16,36] unequivocally confirm their common origin.

Orphan cluster 1 includes a single family of IS*200*/IS*605* transposons which are widespread in bacteria and archaea[37]. The HUH endonucleases of the IS*200*/IS*605* insertion sequences have been extensively studied structurally and biochemically, resulting in a comprehensive understanding of their functions[16,38]. Although IS*200*/IS*605* transposases have a structural fold common to that of other HUH endonucleases and contain all 3 signature motifs, they did not show appreciable sequence similarity to any other cluster of HUH endonucleases and thus remained disconnected from sequences in other clusters. Nevertheless, sequence diversity within the IS*200*/IS*605* cluster is comparable to that within other clusters.

Orphan cluster 2 includes Rep proteins that are conserved in hyperthermophilic archaeal viruses of the family *Rudiviridae*[39]. Structural studies of the Rep protein from the rudivirus SIRV1 revealed the canonical HUH endonuclease fold and biochemical characterization of the protein confirmed the expected nicking and joining activities in vitro[36]. Like the IS*200*/IS*605* transposases, the rudiviral Rep cluster does not connect to other HUH endonucleases, including homologs from other families of archaeal viruses and plasmids.

Conceivably, the uniqueness of the 2 orphan clusters is linked to the unusual transposition and replication mechanisms

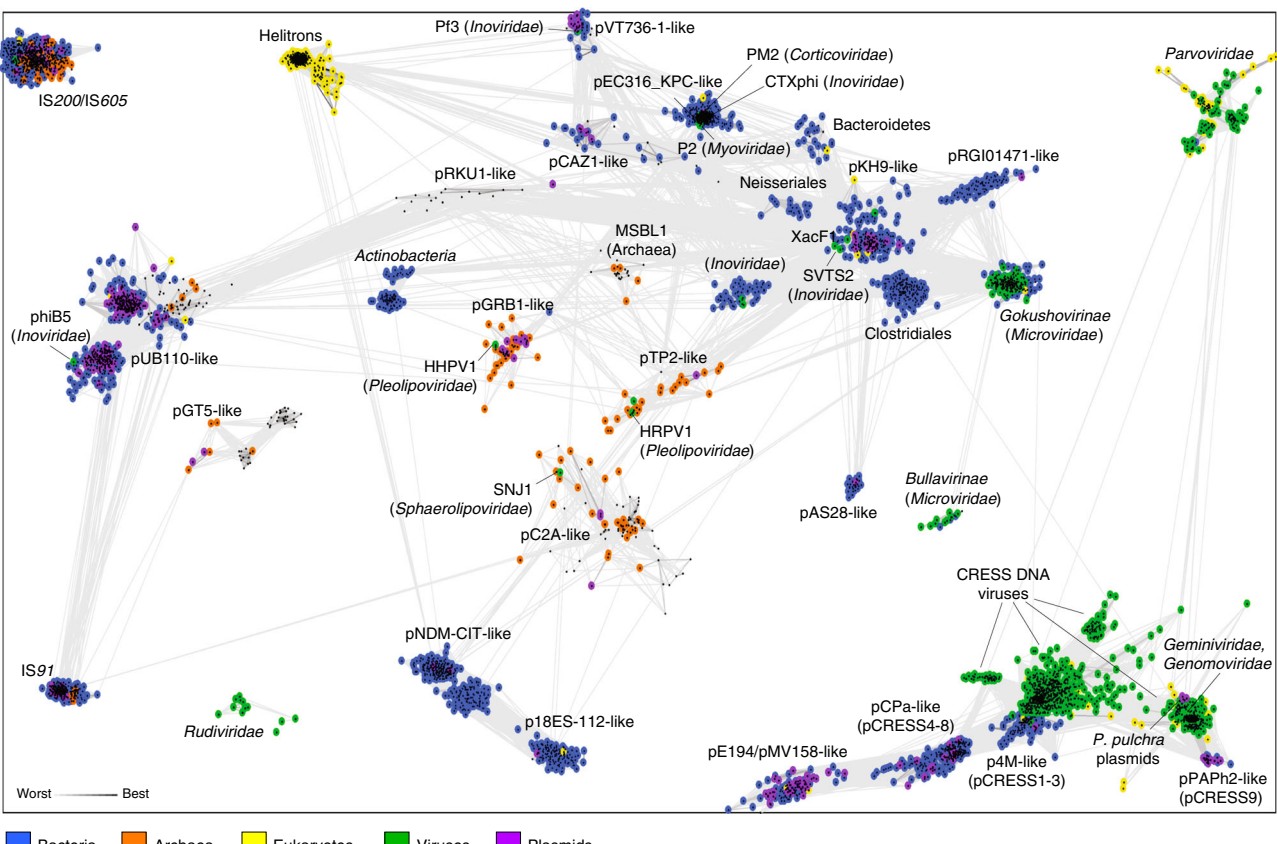

**Fig. 1** Representative HUH superfamily Reps clustered by their pairwise sequence similarity. Lines connect sequences with *P*-value ≤ 1e−08. Groups were named after well-characterized plasmids, viruses or most frequent taxon

employed by the respective elements. Indeed, IS*200*/IS*605* insertion sequences transpose by a unique peel-and-paste mechanism[38], whereas rudiviruses, unlike most other viruses and plasmids replicating by the rolling-circle mechanism, contain relatively large (~35 kb) linear dsDNA genomes with covalently closed termini[40].

Supercluster 1 is by far the largest and most diverse HUH assemblage that includes 24 clusters (Supplementary data 1). Of these 24 clusters, 15 contain Reps from bona fide extrachromosomal plasmids of which 7 clusters also include Reps from diverse ssDNA (*Microviridae*, *Inoviridae*, and *Pleolipoviridae*) and/or dsDNA (*Myoviridae* and *Corticoviridae*) viruses of bacteria and archaea. Three clusters consist of Reps encoded by microviruses of the subfamilies *Gokushovirinae* and *Bullavirinae*, and Xanthomonas inovirus Cf1 (family *Inoviridae*), respectively. Notably, phiX174-like microviruses (*Bullavirinae*) display similarity exclusively to microviruses of the subfamily *Gokushovirinae*, indicative of the Rep monophyly in the two subfamilies of the *Microviridae*, despite high sequence divergence. The bacterial IS*91* (including IS*CR* subfamily) and eukaryotic Helitron family transposons, respectively, form two distinct clusters. The two groups of transposons are not directly connected to each other, but are linked to distinct groups of bacterial and, in the case of IS*91*, archaeal plasmids, suggesting independent origins from bacterial extrachromosomal replicons. It has been previously suggested that helitrons might represent a missing link between eukaryotic CRESS-DNA viruses, namely, geminiviruses, and bacterial HUH replicons[41] or that helitrons evolved from geminiviruses[42]. However, in our analysis, helitrons do not connect to any of the groups of CRESS-DNA viruses, suggesting independent evolutionary trajectories, consistent with the recent findings[43].

The remaining 5 clusters do not include any recognizable plasmid, viral or transposon sequences and thus are likely to represent new families of integrated MGE. Four of these groups are predominantly found in bacteria of the taxa Clostridiales, Actinobacteria, Neisseriales, and Bacteroidetes, respectively (labeled accordingly in Fig. 1), whereas the fifth group is specific to the candidate division MSBL1 (Mediterranean Sea Brine Lakes 1)[44], a group of uncultured archaea found in different hypersaline environments. Most of the clusters display taxonomic uniformity at the domain level, i.e., clusters included either bacterial, or archaeal, or eukaryotic sequences (including the corresponding viruses and plasmids), suggesting that horizontal transfers of viruses or plasmids between host domains are infrequent. The two exceptions include the pUB110-like and IS*91*-like bacteria-dominated clusters, which include a handful of archaeal sequences. In the case of IS*91* transposons, horizontal transfer from bacteria has been ascertained by phylogenetic analyses[45]. In addition, some of the clusters include sporadic sequences annotated as being eukaryotic; however, analysis of the corresponding contigs suggests that these are likely bacterial contaminants.

Of particular interest are the 7 clusters that include both viruses and plasmids. For instance, pEC316_KPC-like cluster, besides plasmids, contains evolutionarily-unrelated viruses from 3 families, *Myoviridae*, *Corticoviridae*, and *Inoviridae*, suggesting extensive horizontal spread of the *rep* genes. Notably, Reps of inoviruses are distributed among 5 clusters. Given the scarcity of inoviral sequences in the pVT736-1-like and pUB110-like clusters, which include only Pseudomonas phage Pf3 and Propionibacterium phage B5, respectively, the directionality of gene transfer, from plasmids to the corresponding viruses,

appears obvious. Furthermore, many inoviruses do not encode HUH endonucleases, but rather encode replication initiators of an evolutionarily unrelated superfamily, *Rep_trans* (Pfam id: PF02486)[15], which also abounds in bacterial plasmids[30], whereas inoviruses of the genus *Vespertiliovirus* lack Reps and instead replicate by transposition using IS*3* and IS*30* family transposases derived from the corresponding insertion sequences[46]. Collectively, these observations indicate that the replication modules of inoviruses have been exchanged with distantly related and even non-homologous replication modules from various plasmid and transposon families. Similarly, archaeal pleolipoviruses are split between two clusters corresponding to different families of archaeal plasmids, pGRB1-like and pTP2-like, respectively, suggesting that exchange of replication-associated genes is common in bacterial and archaeal viruses with small, plasmid-sized genomes. ¶In some cases, it is difficult to ascertain the viral versus plasmid membership of Reps encoded in cellular chromosomes because both types of MGE can integrate into the host genomes. For example, the XacF1-like cluster includes 62 Rep sequences, 2 of which are encoded by filamentous phages, whereas the rest come from bacterial genomes. Analysis of the genomic neighborhoods suggests that only 6 of the remaining 60 Reps represent prophages. Furthermore, the pAS28-like cluster includes one plasmid, pAS28 (ref. [47]); however, related Reps have been previously identified in prophages[48], but not in characterized viruses, giving the erroneous impression that the pAS28-like Rep is plasmid-exclusive. ¶To further characterize the evolutionary relationships between Reps encoded by different types of MGE, we constructed maximum likelihood phylogenetic trees for the 7 clusters that included Reps from both viruses and plasmids (Supplementary Fig. 2a-g). The results of phylogenetic analyses suggest horizontal transfer of the *rep* genes between plasmids and viruses, with viral sequences typically being nested among plasmid-encoded homologs.

Supercluster 2 (SC2) consists of 7 clusters (Supplementary data 1) which include all known classified and unclassified eukaryotic CRESS-DNA viruses, parvoviruses, a cluster of plasmids from the red alga *Pyropia pulchra*[49], and 4 clusters containing bacterial Rep sequences. The vast majority of the bacterial Reps in the pCPa-like and p4M-like clusters are encoded in bacterial genomes rather than in plasmids and have not been previously characterized. In our network, the CRESS-DNA viruses are connected to pCPa-like, p4M-like, pPAPh2-like and *P. pulchra*-like clusters, whereas the pE194/pMV158-like cluster does not form direct connections to the CRESS-DNA viruses, but joins SC2 through the pCPa-like cluster (Fig. 1). Notably, geminiviruses and genomoviruses form a subcluster with plasmids of phytoplasma (pPAPh2-like cluster) and *P. pulchra*, which is separated from other CRESS-DNA viruses. The *Parvoviridae* cluster, including parvoviruses and derived endogenous viruses integrated in various eukaryotic genomes, is loosely connected directly to the CRESS-DNA viruses, suggesting that parvoviruses with linear ssDNA genomes share common ancestry with CRESS-DNA viruses which, by definition, have circular genomes. Intrigued by the seemingly close evolutionary connection between eukaryotic CRESS-DNA viruses and bacterial and algal Reps, we investigated these relationships in greater detail, as reported in the following sections.

**The diversity of viral-like Reps in bacterial genomes**. To investigate the extent of similarity between the Reps of eukaryotic CRESS-DNA viruses and non-viral replicons from SC2, we compared their domain organizations. With the exception of pE194/pMV158-family plasmids, which contain only the nuclease domain, bacterial and algal SC2 Reps had the same nuclease-helicase domain organization as CRESS-DNA viruses. The same two-domain organization is also characteristic of the parvovirus Reps[2]. Thus, domain organization analysis corroborates the results of sequence clustering and further indicates that the bacterial SC2 Reps are more closely related to the Reps of eukaryotic viruses than to those from other prokaryotic plasmids and viruses.

We then sought to obtain additional information on the diversity and taxonomic distribution of the viral-like SC2 Reps that are encoded in bacterial genomes. Maximum likelihood phylogenetic analysis revealed 9 well-supported clades (Fig. 2a). Clustering and subsequent community detection analysis validated the 9 groups of bacterial Reps (Fig. 2b), where groups 1–3 correspond to the p4M-like cluster shown in Fig. 1, groups 4–8 to the pCPa-like cluster, and group 9 to the pPAPh2-like cluster. To emphasize their similarity to Reps of CRESS-DNA viruses, we refer to the 9 groups as pCRESS1 through pCRESS9. These groups displayed partially overlapping but distinct taxonomic distributions, covering several classes within 4 bacterial phyla (Supplementary Fig. 1 and Supplementary Table 1).

The majority of the Reps from pCRESS7 and pCRESS9 are encoded by extrachromosomal plasmids (Supplementary Table 1). By contrast, the vast majority (97.5%) of Reps found in other groups are encoded within mobile genetic elements site-specifically integrated into bacterial chromosomes (Supplementary Table 1; Fig. 2c; Supplementary Fig. 3; Supplementary Note 1). Notably, none of the elements encoded any homologs of currently known viral structural proteins (Supplementary Note 1). Collectively, these observations indicate that viral-like Reps in bacteria are encoded by diverse extrachromosomal and integrated plasmids.

**Conserved features of bacterial and CRESS-DNA virus Reps**. Sequence analysis showed that, despite considerable overall sequence divergence, Reps of pCRESS4 through 8 contain closely similar sequence motifs within the nuclease and helicase domains (Fig. 3), consistent with the results of the clustering and phylogenetic analyses (Fig. 2). In particular, these 5 pCRESS groups share a specific signature, YLxH (x, any amino acid) within motif III of the nuclease domain, which was not observed in Reps from pCRESS1–3 and 9 (Fig. 3). Thus, we refer to pCRESS4–8 collectively as the YLxH supergroup (rather than the pCPa-like cluster), to emphasize this shared feature. The YLxH signature was also conserved in Reps from the pE194/pMV158-like cluster, suggesting a closer evolutionary relationship between the two clusters, despite the fact that pE194/pMV158-like Reps lack the helicase domain. Also, pCRESS9 displays motifs similar to those of the *P. pulchra* plasmids and thus could be unified with these plasmids into a common assemblage. By contrast, pCRESS1, -2 and -3 (p4M-like cluster) display distinctive sets of motifs (Fig. 3; Supplementary Note 1).

**Origin of the SF3 helicase domain**. Sequence analyses suggest that the SF3 helicase domain-containing plasmid Reps, especially those from pCRESS2, pCRESS3, and pCRESS9, and *P. pulchra*, are closely related to the Reps of CRESS-DNA viruses. However, the directionality of evolution, i.e., whether plasmid Reps evolved from those of CRESS-DNA viruses or vice versa, is not obvious. Although it is tempting to take the absence of the helicase domain in the pE194/pMV158-like cluster as an indication that this group is ancestral to the helicase-containing Reps, it cannot be ruled out that the helicase domain was lost by these plasmids. Thus, we set out to investigate the provenance of the SF3 helicase domain in the plasmid and viral Reps. Sensitive sequence searches with HMMER against the nr30 database showed that the helicase

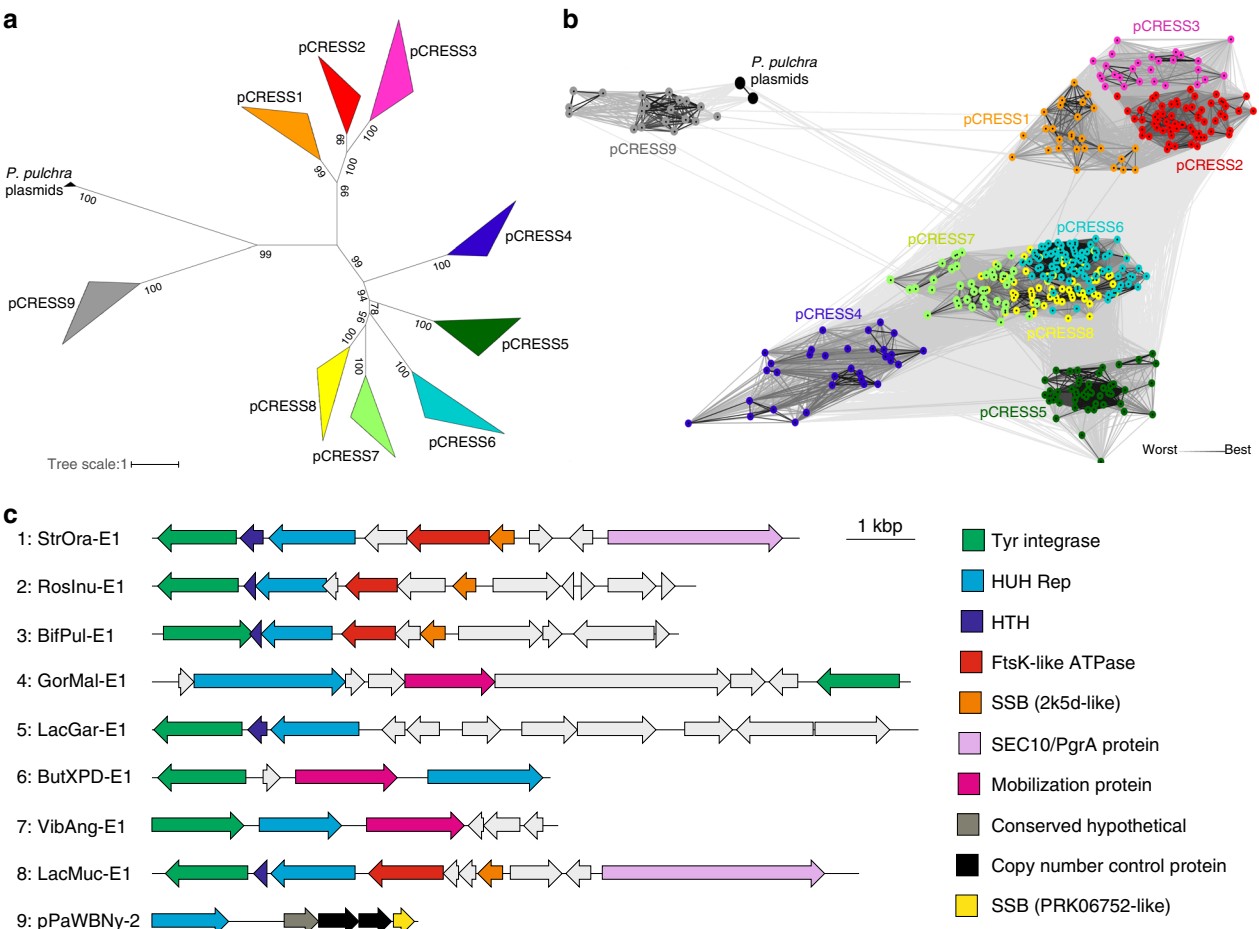

**Fig. 2** Diversity of viral-like Rep proteins in bacteria. **a** Phylogenetic tree of bacterial Rep proteins and their homologs in *P. pulchra*. Closely related sequences are collapsed to triangles, whose side lengths are proportional to the distances between closest and farthest leaf nodes. **b** CLANS groups of bacterial Rep proteins and their homologs. Nodes indicate protein sequences. Lines represent sequence relationships (CLANS *P*-value ≤ 1e−05). The nodes belonging to the same cluster are colored with the same colors, corresponding to the clades shown in panel A. **c** Genome maps of integrated and extrachromosomal plasmids representing groups 1–9. Homologous genes are depicted using the same color and their functions are listed on the right side of the figure

domains of plasmid and CRESS-DNA viral Reps are most closely related to those of eukaryotic positive-sense RNA viruses (order *Picornavirales* and family *Caliciviridae*) as well as the AAA+ ATPase superfamily[50,51]. In this analysis, we also included the SF3 sequences of parvoviruses, polyomaviruses, and papillomaviruses that are thought to be evolutionarily related to CRESS-DNA viruses[2,25]. Several groups of more distant SF3 helicases from viruses with large dsDNA genomes[52] were disregarded. Due to the high sequence divergence and relatively short length, phylogenetic analyses of the SF3 helicase domains were not informative, resulting in star-shaped tree topologies, irrespective of the evolutionary models or taxonomic sampling used. However, clustering analysis based on pairwise similarities provided insights into the relationships between the different ATPase families (Fig. 4a). In particular, the close relationship between the SF3 helicase domains of bacterial Reps and CRESS-DNA viruses was clearly supported. Both groups connect to the RNA viruses, but only bacterial Reps, particularly those of the YLxH supergroup, show connections to AAA+ superfamily ATPases, namely, bacterial helicase loader DnaC and, to a lesser extent, DnaA and Cdc48-like ATPases (Fig. 4a). The closer similarity between the YLxH supergroup and bacterial AAA+ ATPases is supported by comparison of the catalytic motifs which revealed several shared derived characters, to the exclusion of other groups

(Supplementary Fig. 4). At the same clustering threshold, neither eukaryotic DNA nor RNA viruses linked to any group of ATPases other than those from bacterial plasmids. The SF3 helicases of parvoviruses linked to those of CRESS-DNA viruses, consistent with the analysis of full-length Rep sequences (Fig. 1). Papillomaviruses and polyomaviruses formed 2 clusters which connected to each other and to parvoviruses.

This pattern of connectivity suggests a specific vector of evolution and appears to be best compatible with the following scenario. The SF3 helicase domain of bacterial plasmids evolved from a bacterial DnaC-like ATPase; this helicase domain was appended to the nuclease domain of Reps of pE194/pMV158-like plasmids yielding the ancestor of the YLxH supergroup; bacterial plasmid Reps were passed on to the CRESS-DNA viruses; the SF3 helicase of RNA viruses was horizontally acquired either from bacterial plasmids or, more likely, from eukaryotic CRESS-DNA viruses; CRESS-DNA viruses have spawned parvoviruses which in turn gave rise to polyomaviruses and papillomaviruses (Fig. 4b). The alternative scenario, under which SF3 helicases of eukaryotic RNA viruses gave rise to the universal bacterial DnaC and DnaA proteins, through bacterial plasmids, appears non-parsimonious and extremely unlikely. Indeed, DnaA is ubiquitous and essential in bacteria[50,51], so the capture of the helicase from a plasmid would have to occur at the very origin of the bacterial domain of

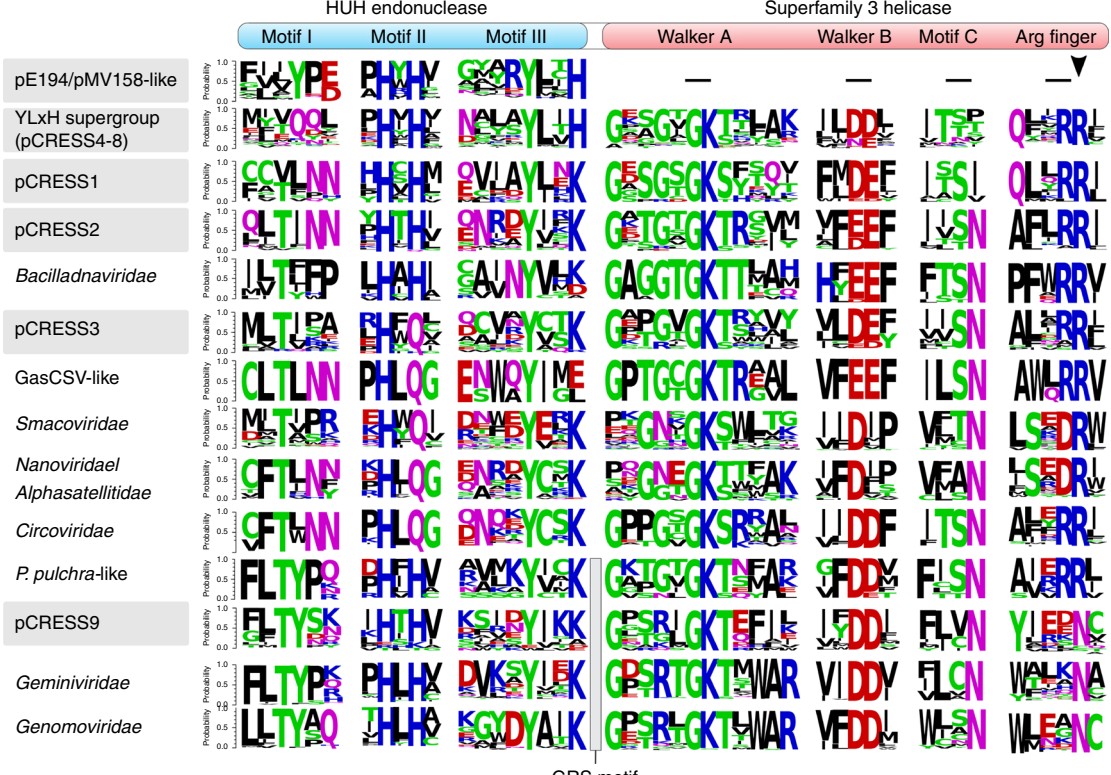

**Fig. 3** Conserved sequence motifs of Rep proteins. Bacterial Rep groups are depicted in gray background. Residues are colored by their chemical properties (polar, green; basic, blue; acidic, red; hydrophobic, black; neutral, purple). The Rep groups were manually ordered according to the pairwise similarity in the aligned motifs. The HUH endonuclease and SF3 helicase domains are delineated at the top of the figure

life. Notably, pCRESS9 and *P. pulchra* plasmids are not linked with other plasmids but are rather connected to the rest of the sequences through the CRESS-DNA viruses. The latter pattern has been also observed in the global clustering analysis of the HUH Reps (Fig. 1) as well as in the clustering of the nuclease domains alone.

**Origins of CRESS-DNA viruses from bacterial plasmids.** Analysis of the SF3 helicase domains suggests that Reps of pE194/ pMV158-like plasmids are ancestral rather than derived forms. The alternative possibility, namely, that Reps of pE194/pMV158-like plasmids have lost the helicase domain, cannot be currently ruled out. However, the fact that the helicase domain has not been lost in any of the numerous known groups of CRESS-DNA viruses or in pCRESS1 to pCRESS9 plasmids, suggests that, once acquired, the helicase domain becomes important for efficient plasmid/viral genome replication. Thus, the close similarity between the pE194/pMV158-like Reps and those of the YLxH supergroup, resulting in direct connectivity of the two groups in the global network (Fig. 1), implies that the former group is an adequate outgroup for the phylogeny of Reps from bacterial plasmids and CRESS-DNA viruses. For phylogenetic analyses, we used a dataset of SC2 Reps, excluding Reps of *Parvoviridae* and CRESS-DNA viruses which were previously judged to be chimeric with respect to their nuclease and helicase domains[53], to avoid potential artifacts resulting from conflicting phylogenetic signals. The dataset included representatives of all classified families of CRESS-DNA viruses as well as 6 groups of unclassified CRESS-DNA viruses provisionally labeled CRESSV1–6 (ref. [53]) as well as a small group of GasCSV-like viruses, which have been previously noticed to encode Reps with significant similarity to bacterial Reps[54]. In the well-supported maximum likelihood phylogenetic

tree constructed with PhyML and rooted with pE194/pMV158-like Reps, the YLxH supergroup (pCRESS4–8) is at the base of an assemblage that includes all CRESS-DNA viruses, pCRESS1–3 and pCRESS9 as well as *P. pulchra* plasmids. This assemblage splits into two clades (Fig. 5). Clade 1 includes two subclades, one of which consists of geminiviruses and genomoviruses joining pCRESS9 plasmids of phytoplasma, and the other one includes CRESSV6 and *P. pulchra* plasmids. Notably, *P. pulchra* plasmids appear to emerge directly from within the CRESSV6 diversity, with the closest relationship to the CRESSV6 subclade of viruses sequenced from wastewater samples. The relationship between geminiviruses/genomoviruses and pCRESS9 plasmids is not resolved in the phylogeny. However, clustering analyses strongly suggest that Reps of pCRESS9 plasmids evolved from geminiviruses-genomoviruses (Figs. 1 and 4). Consistent with this scenario, phytoplasmal pCRESS7 and pCRESS9 plasmids, despite encoding phylogenetically distinct Reps, share the gene content, namely, the copy number control protein, PRK06752-like SSB protein and conserved hypothetical protein (Supplementary Fig. 3g, i). Furthermore, geminiviruses and CRESSV6 encode homologous capsid proteins suggesting that they evolved from a common viral ancestor rather than converged from two groups of plasmids by capturing homologous capsid protein genes. Clade 2 includes bacterial Reps of pCRESS1–3 and, as a sister group, CRESS-DNA viruses of the families *Nanoviridae/Alphasatelliti-dae*, *Smacoviridae*, and *Circoviridae* as well as unclassified CRESSV1 through CRESSV5, whereas GasCSV-like viruses are nested within bacterial pCRESS2.

The robustness of the PhyML tree was validated by additional analyses (Supplementary Note 1), including (i) maximum likelihood phylogenetic analyses using RAxML and IQ-Tree, with alternative branch support methods (Figure S5); (ii) phylogenetic reconstruction using the 20-profile mixture model (Figure S5);

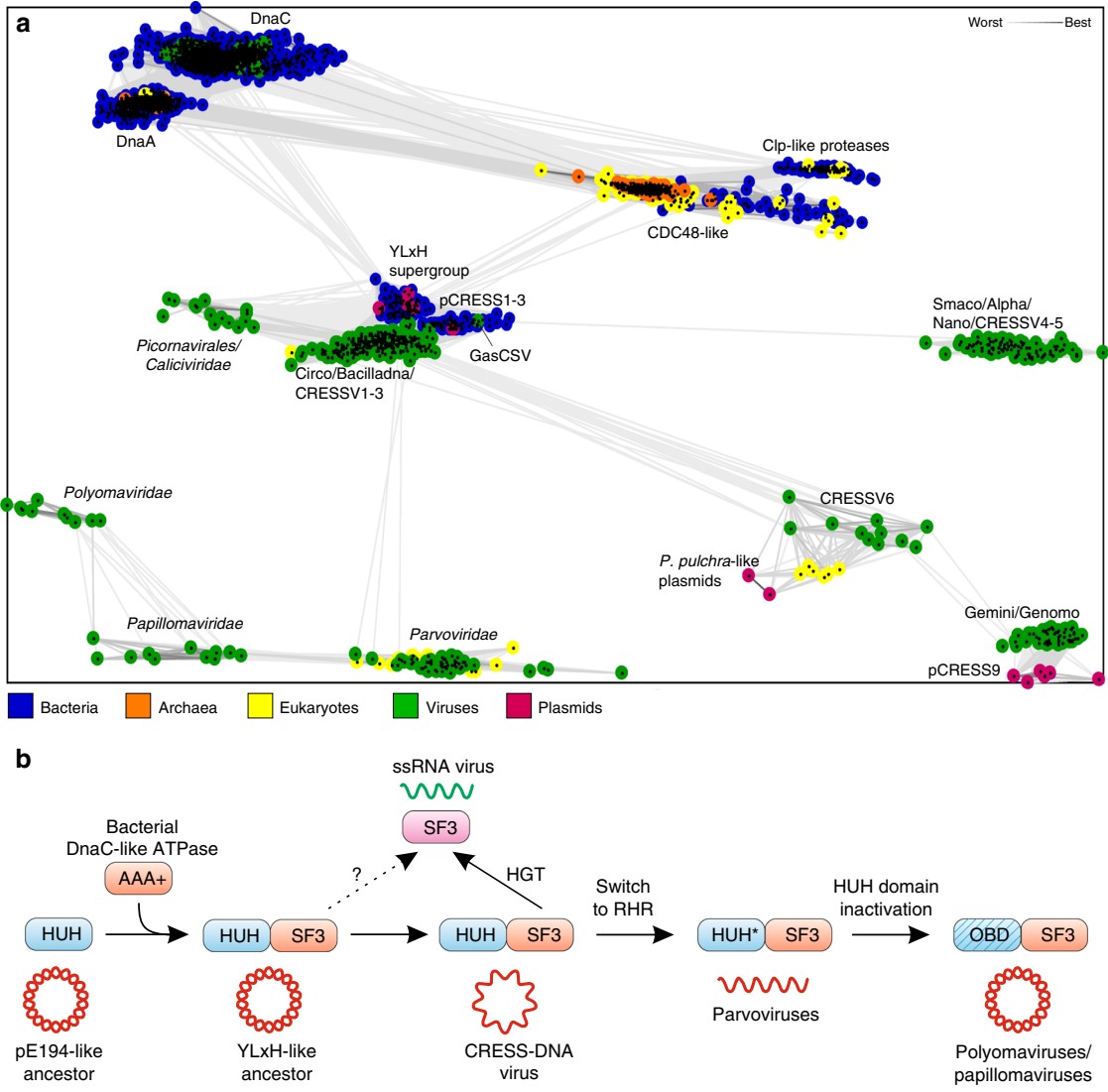

**Fig. 4** Relationships between Superfamily 3 helicases and AAA+ ATPases. **a** Superfamily 3 helicase and AAA+ ATPase domains clustered by their pairwise similarity using CLANS. In total, 3854 sequences were clustered with CLANS (CLANS *P*-value ≤ 5e−09). Groups of unclassified CRESS-DNA viruses are referred to as CRESSV1 through CRESSV6 (ref. [53]). **b** A proposed evolutionary scenario for the origin and evolution of viral Superfamily 3 helicases. Abbreviations: SF3, superfamily 3 helicase domain; HUH, HUH superfamily nuclease domain; OBD, origin-binding domain; HGT, horizontal gene transfer; RHR, rolling-hairpin replication

(iii) statistical analysis of the unconstrained and 3 constrained tree topologies (Supplementary Table 2). Collectively, these results indicate that the obtained tree topology is highly robust and is likely to accurately reflect the evolutionary history of Reps encoded by CRESS-DNA viruses and plasmids.

Notably, analysis of the conserved motifs (Fig. 3) suggests a specific association between the virus Reps in clade 1 and bacterial pCRESS3 (rather than pCRESS1–3 collectively), implying that the phylogenetic placement might be affected by ancient recombination events. Furthermore, bacilladnaviruses were omitted from the global phylogenetic tree because their Reps displayed unstable position in the phylogeny depending on the taxon sampling (Supplementary Fig. 6), possibly, due to the small number of available sequences, their high divergence and potential chimerism. Regardless, phylogenetic analysis strongly suggests that the majority of CRESS-DNA viruses, including circoviruses, smacoviruses, nanoviruses, and CRESSV1–5, evolved from a common ancestor with bacterial Reps of pCRESS1–3, whereas the uncultivated GasCSV-like viruses emerge directly from the bacterial pCRESS2 Reps (Fig. 5). The

provenance of the assemblage including geminiviruses, genomoviruses and CRESSV6 is less clear but might predate the emergence of the other CRESS-DNA virus groups and possibly involved a common ancestor with the YLxH supergroup. The Reps of bacterial pCRESS9 and *P. pulchra* plasmids have been likely horizontally acquired more recently from the corresponding CRESS-DNA viruses.

## Discussion

Here, we explored the evolutionary relationships among different classes of bacterial, archaeal, and eukaryotic replicons encoding HUH endonucleases (Reps). Our analysis revealed widespread exchange of *rep* genes among bacterial and archaeal viruses and plasmids, with some of the Rep clusters being particularly promiscuous, as in the case of pEC316_KPC-like Reps which are encoded not only in the corresponding plasmid but also in evolutionarily unrelated bacteriophages from 3 different families. Conversely, Reps of filamentous bacteriophages (family *Inoviridae*) fall into 5 distinct HUH clusters, indicating that, in this

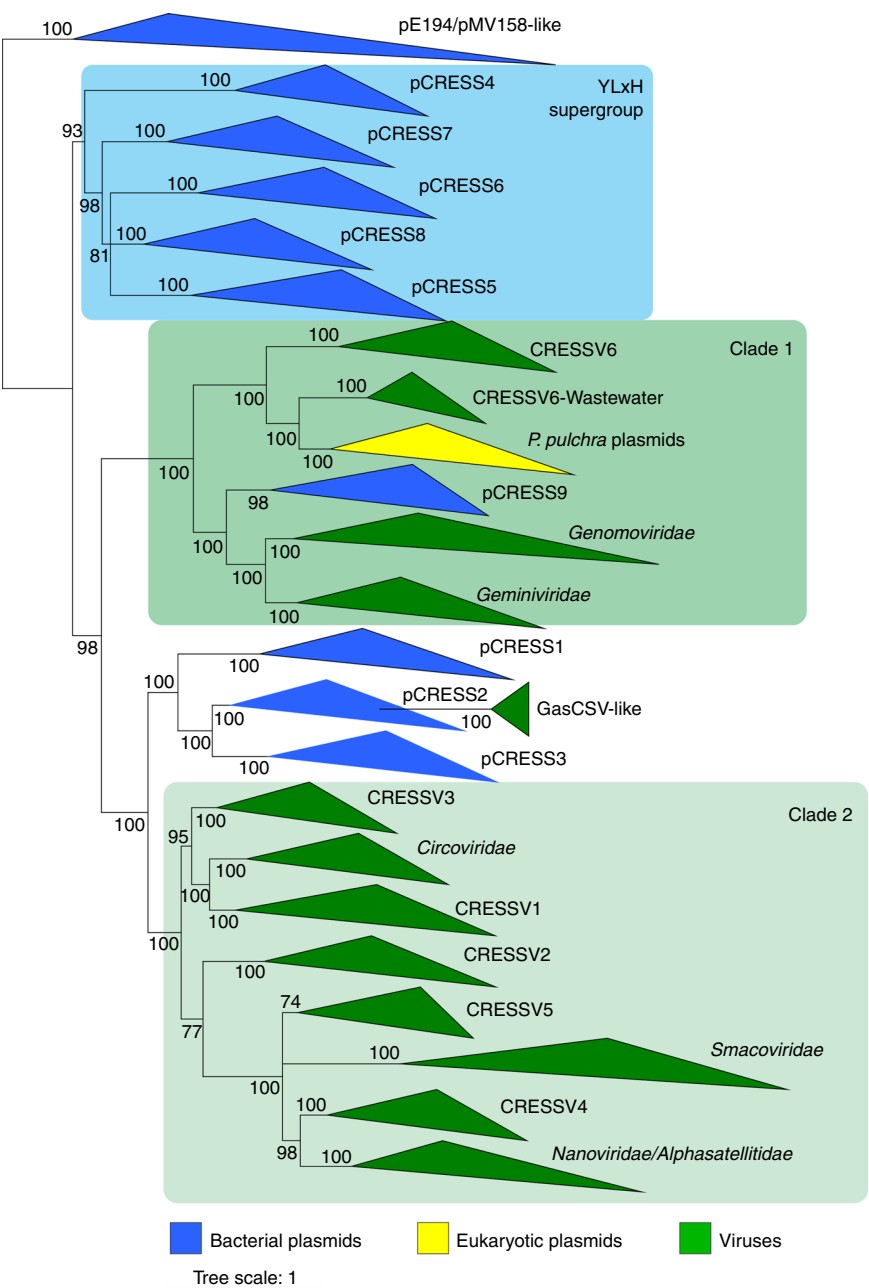

**Fig. 5** Maximum likelihood phylogenetic tree of Rep proteins. GasCSV—Gastropod-associated circular ssDNA virus. The tree was constructed with PhyML[78]. Branches with support values below 70 are contracted

virus family, replication modules are readily exchangeable, presumably, for those better suited in particular hosts. Thus, the genome replication module of inoviruses shows extreme promiscuity and cannot serve as a phylogenetic marker, consistent with a recent analysis of 10,000 inovirus genomes[15], so that the family is held together by the shared morphogenetic module. By contrast, the *rep* is the only gene conserved in all CRESS-DNA viruses and can serve as a vertically transmitted character against which various evolutionary events associated with the diversification of this virus class are mapped. Ultimately, however, no single gene or even functional module can fully represent the evolution of a given virus group. Instead, a more "holistic" approach is needed, where the provenance of all or most virus genes is deciphered.

Our present analysis pinpoints the origins of the replication modules of CRESS-DNA viruses. We identified 9 groups of

bacterial Reps which share the nuclease-helicase domain organization with CRESS-DNA viruses. These bacterial Reps are encoded by previously unknown plasmids integrated into the genomes of diverse bacteria. By tracing the evolution of the helicase domain, we inferred the likely vector of evolution, namely, from plasmid Reps to the Reps of CRESS-DNA viruses. Although the Reps of CRESS-DNA viruses are generally considered to be monophyletic[17], our analysis shows that this might not be the case *sensu stricto*. Instead, the CRESS-DNA virus diversity has likely been seeded on 3 independent occasions from different groups of bacterial plasmids at different stages of evolution (Fig. 6). Conversely and contrary to the previous conclusion[33], our results also indicate that CRESS-DNA viruses have given rise to (at least) 2 groups of plasmids in red alga and phytopathogenic phytoplasma, respectively. Thus, transitions between the virus and plasmid states appear to be bidirectional.

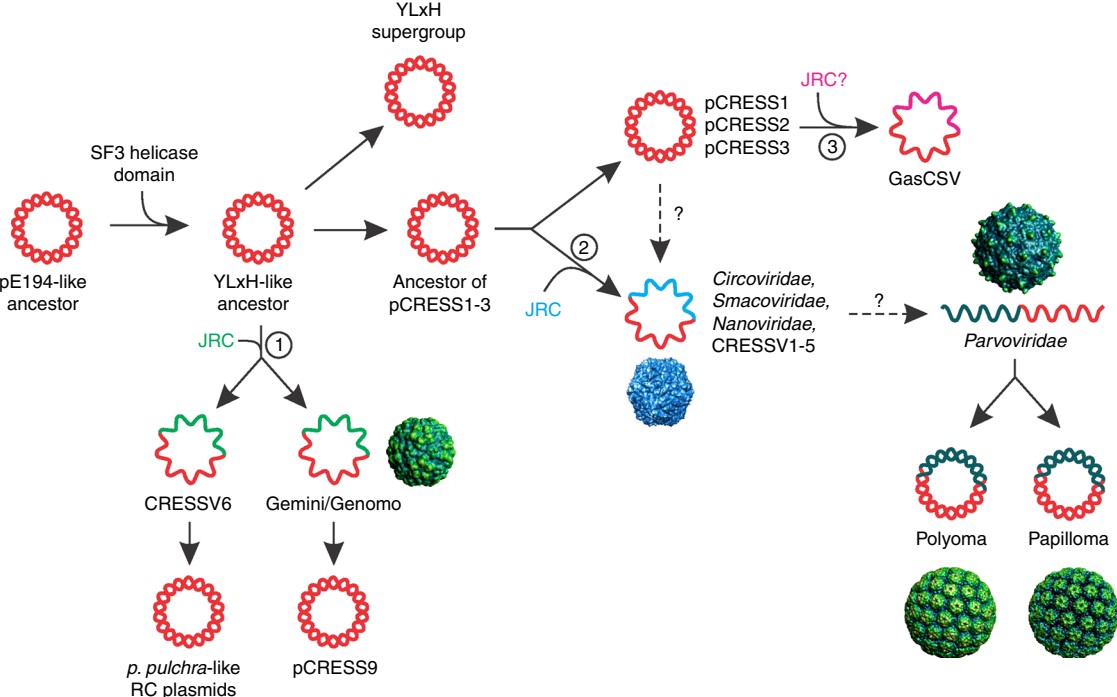

**Fig. 6** A proposed evolutionary scenario for the origin of CRESS-DNA viruses from bacterial plasmids. The three hypothetical events of CRESS-DNA virus emergence are indicated with numbered circles. JRC, jelly-roll capsid protein-encoding genes. Non-orthologous JRC genes are indicated with different colors

Obviously, transformation of a plasmid into a virus involves acquisition of the morphogenetic module, i.e., minimally, a gene for the capsid protein. We and others have previously shown that capsid proteins of different groups of CRESS-DNA viruses display specific relationships to single jelly-roll capsid proteins of RNA viruses of different families[33,55–58]. Thus, we propose that CRESS-DNA viruses evolved from plasmids through acquisition of reverse-transcribed capsid protein genes from different groups of RNA viruses (Fig. 6). It seems likely that the capture of the capsid protein genes from RNA viruses has occurred in eukaryotic cells, possibly involving symbiotic or parasitic bacterial donors of the corresponding plasmids. Some of these events could have occurred at the early stages of eukaryotic evolution, as in the case of the viral assemblage including circoviruses, smacoviruses, nanoviruses and CRESSV1–5. By contrast, GasCSV-like viruses probably emerged relatively recently. Given the close relationship between pCRESS2 and GasCSV-like viruses, viruses of the latter group might infect bacteria rather than eukaryotes. Alternatively, the transition from a plasmid to a CRESS-DNA virus ancestor could have occurred once and was followed by replacement of the *rep* genes with counterparts from other plasmids, resulting in the 3 contemporary lineages of CRESS-DNA viruses. However, given that neither Rep nor capsid proteins appear to be orthologous in the 3 virus groups, this alternative scenario cannot be substantiated at this point. Regardless, it is clear that Rep and capsid protein genes have been repeatedly exchanged with distantly related homologs from other viruses, even in the more recent history of CRESS-DNA viruses[55,58,59]. Notably, it has been recently suggested based on the presence of matching CRISPR spacers that smacoviruses infect methanogenic archaea[60]. In our phylogeny (Fig. 5), smacoviruses are deeply nested among circoviruses and nanoviruses, for which eukaryotic hosts have been confirmed experimentally[1]. Thus, if smacoviruses are shown to infect archaea as recently suggested[60], the phylogeny is best compatible with a eukaryote to prokaryote transfer. The hosts of CRESSV1–6 are currently unknown and might include organisms from any of the 3 cellular domains of life. Furthermore, given that

the SF3 helicase domain is now found in Reps of diverse bacterial replicons, this signature should be considered with caution when attributing viral genomes discovered by metagenomics to particular hosts.

Our findings further suggest that parvoviruses that have linear ssDNA genomes evolved directly from CRESS-DNA viruses. Indeed, both Rep and capsid proteins of parvoviruses are homologous to those of the CRESS-DNA viruses[2]. Unlike the Reps involved in rolling-circle replication, the Rep of parvoviruses lacks the joining activity used by CRESS-DNA viruses to circularize progeny genomes. Instead, the parvovirus Rep remains covalently attached to the 5′ ends of all viral DNA molecules[20]. The eukaryotic viruses with small, circular dsDNA genomes that comprise the families *Polyomaviridae* and *Papillomaviridae* encode major replication proteins that consist of an SF3 helicase domain and an inactivated HUH nuclease domain, lacking all 3 signature motifs. Nevertheless, structural studies have unequivocally demonstrated that the N-terminal origin-binding domains of both polyomaviruses and papillomaviruses are homologous to the HUH endonuclease domains of CRESS-DNA viruses and parvoviruses[61]. Thus, these viruses, most likely, evolved from ssDNA viruses but their evolution involved a drastic change in both the genome DNA structure and the replication mechanism such that the HUH domain switched from an enzymatic to a structural role. Clustering analysis of the SF3 helicase domains suggests that both polyomaviruses and papillomaviruses evolved from parvoviruses, although the driving forces behind this transition remains obscure.

The current classification of CRESS-DNA viruses largely relies on the phylogeny of the Rep proteins[17,62,63]. The ever-growing diversity of sequenced CRESS-DNA virus genomes calls for revision of the taxonomy of this class of viruses. Our analysis reveals two larger groupings of CRESS-DNA viruses, each including several families/clades, which could be equivalent to new orders, whereas all CRESS-DNA viruses could be unified at a yet higher taxonomic level. Finally, the membership of parvoviruses,

polyomaviruses, and papillomaviruses in this assemblage could be also considered, at the highest taxonomic level. Indeed, it is not unprecedented that the same taxon contains viruses with different nucleic acids types. For instance, the order *Ortervirales* includes reverse-transcribing viruses with RNA and DNA genomes[64], whereas members of the family *Pleolipoviridae* have either ssDNA or dsDNA genomes[65]. Notably, the ICTV has recently announced that taxonomic ranks above the order level are now officially accepted[66], opening the door for the formal unification of the whole spectrum of evolutionarily related CRESS-DNA viruses.

Although the Reps of prokaryotic viruses of the families *Microviridae*, *Inoviridae,* and *Sphaerolipoviridae* lack the helicase domain, their HUH nucleases show clear affinities with those from different groups of plasmids, suggesting routes of evolution parallel to those of the CRESS-DNA viruses. Furthermore, the evolution of inoviruses appears to have involved multiple replacements of the *rep* gene with those from various plasmids.

The results presented here shed light on the origin of a major part of the virosphere, the ssDNA viruses replicating via the rolling-circle mechanism, and in particular, CRESS-DNA viruses. Arguably, evolution of the ssDNA viruses is the most compelling manifestation of the previously noted general trend in virus evolution, namely, tight evolutionary connections between viruses and capsid-less MGE[67,68].

## Methods

**Databases**. Homologs of the HUH endonucleases were retrieved by running searches against protein sequence databases filtered to 50 and 90% sequence identity (UniRef50 and UniRef90, respectively) which were downloaded from http://www.uniprot.org. Search for bacterial homologs of CRESS-DNA virus Reps was performed against nr90 (NCBI's nr database (ftp://ftp.ncbi.nlm.nih.gov/blast/db/) filtered to 90% identity). To detect remote sequence similarity, we used sequence profile databases which included profiles from PDB (www.pdb.org), SCOP[69], Pfam[70], and CDD[71]. For query profile generation nr70 database was used.

**Sequence searches and clustering**. Homologs of the HUH superfamily endonuclease domains for each representative Rep sequence were obtained by performing three jackhmmer[72] iterations against the UniRef50 database. Representative Reps were selected as queries for homology searches based on exhaustive review of literature on the HUH superfamily[16,17,28,53]. In addition, for HUH groups with less than 10 homologs in UniRef50, we repeated searches against UniRef90 database. For homology searches only the HUH endonuclease domain was used to avoid attracting unrelated proteins, for example, containing superfamily 1 or 3 helicase domains. However, clustering was performed using full-length sequences to better reflect their evolutionary history. Dataset obtained by searches against the UniRef databases was supplemented with CRESS-DNA virus Reps devoid of obvious recombinant sequences from our previous study[53]. Sequences were clustered using CLANS with BLAST option[35]. CLANS is an implementation of the Fruchterman-Reingold force-directed layout algorithm, which treats protein sequences as point masses in a virtual multidimensional space, in which they attract or repel each other based on the strength of their pairwise similarities (CLANS *p*-values)[35]. Thus, evolutionarily more closely related sequences gravitate to the same parts of the map, forming distinct clusters. Rep clusters were identified by CLANS convex algorithm at *P*-value = 1e−08. To collect bacterial homologs of CRESS-DNA virus Reps, we used representative sequences as queries and performed two jackhmmer iterations against nr90 database. The resultant set of sequences was grouped using a convex clustering algorithm (at *P*-value = 1e−05) in CLANS. To ensure that we gathered all bacterial homologs, HMM profiles were constructed for each identified cluster and used as queries for searches against nr90 with hmmsearch[72]. Accessions of proteins for each group, shown in Fig. 1, are available for download (Supplementary Data 1). For collection of the SF3 helicase dataset, the helicase domain of a YLxH supergroup member from *Streptococcus canis* (WP_003048523) was used as a query for hmmer search against nr30 database available at the Bioinformatics Toolkit server[73]. The resulting dataset was supplemented with SF3 helicase sequences from CRESS-DNA viruses[53], polyomaviruses, papillomaviruses, parvoviruses and *P. pulchra*-like plasmids (Supplementary Data 1). Extracted helicase domains were filtered to 70% identity with CD-HIT (parameter "-c 0.7")[74].

**Remote homology detection**. Sequence searches based on profile-profile comparisons were used to detect remote homology. For profile generation, two iterations of jackhmmer[72] were run against nr70 sequence database using E-value = 1e−03 inclusion threshold. The resulting profiles were used to search against profile databases with HHsearch[75]. Search results for proteins from representative bacterial plasmids and integrative elements are available in Supplementary Data 1.

**Multiple sequence alignments and phylogenetic analysis**. To construct multiple sequence alignments for phylogenetic analysis we used MAFFT[76] and TrimAl[77]. MAFFT options G-INS-i and L-INS-i and TrimAl gap thresholds 0.05 and 0.15 were used to generate alignments for Figs. 2 and 5, respectively. The resulting alignments covered both HUH and SF3 (where available) domains and contained 743 and 508 positions, respectively. Both alignments can be found in the Supplementary Data 2 and 3. Phylogenetic trees were calculated with PhyML[78] using automatic model selection and aBayes branch support. Substitution models VT + G + I + F (VT, amino acid replacement matrix; G, gamma shape parameter: estimated (1.864); I, proportion of invariable sites: estimated (0.005); F, equilibrium frequencies: empirical) and LG + G (LG, amino acid replacement matrix; G: estimated (1.807)) substitution models were selected for phylogenetic analyses shown in Figs. 2 and 5, respectively. Additional trees were constructed using IQ-Tree v1.6.8 (ref. [79]) with Ultrafast Bootstrap Approximation branch support[80], and RAxML with non-parametric bootstrapping[81]. Mixture model tree was constructed with IQ-Tree[79] using model parameters (LG + C20 + F + G) and ultrafast bootstrap (with 1000 replicates). Alignment and guide tree (parameters "-s" and "-ft", respectively) were the same as in Fig. 5. Highly diverged sequences forming long branches were removed before constructing final trees. *Bacilladnaviridae* viruses were also removed, because their position was not stable in trees with different sequence sampling (Supplementary figure 6). Phylogenetic trees are available from the authors upon request. The trees shown in Figs. 2, 5, S5 and S6 can be found in the Supplementary Data 4 to 10.

**Statistical tests**. Alternative topologies for the Rep tree were tested using the IQ-Tree software version 1.6.8 with the following parameters: -m LG+G -n 0 -zb 100000 -zw -au (ref. [79]). As an unconstrained tree, we used the original PhyML tree (Fig. 5), which was tested against each of the constrained trees. The following tests were performed: Approximately Unbiased (AU) test[82], logL difference from the maximal logl in the set, RELL test[83], one sided and weighted Kishino–Hasegawa (KH) tests[84], Shimodaira–Hasegawa (SH) test[85], weighted SH test, Expected Likelihood Weight (ELW) test[86].

**Sequence logos**. Sequence logos for the Reps of CRESS-DNA virus families were taken from ref. [57]. Alignments for other groups were obtained from an alignment used to build the tree shown in Fig. 5. Sequence logos were created using WebLogo server[87].

**Genomic context analysis**. The integrated plasmids were identified by thorough analysis of genomic neighborhoods of the Rep-encoding genes. The precise borders of integration were defined based on the presence of direct repeats corresponding to attachment sites. The repeats were searched for using Unipro UGENE[88]. Genes of integrated plasmids were annotated based on the HHsearch searches[75]. Genome maps were compared and visualized using Easyfig with tBLASTx option[89].

**Reporting summary**. Further information on research design is available in the Nature Research Reporting Summary linked to this article.

## Data availability

The authors declare that the data supporting the findings of this study are available within the paper and its supplementary information files. Accession numbers for all proteins analyzed in this study as well as alignments used to generate the trees shown in Figs. 2 and 5 are included in the Supplementary Data 1, 2, and 3.

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

## Acknowledgements

The authors are grateful to Valerian V. Dolja for insightful comments on the manuscript. M.K. was supported by l'Agence Nationale de la Recherche (project ENVIRA, #ANR-17-CE15–0005-01). D.K. was partly supported by a Short Term Fellowship from the Federation of European Biochemical Societies (FEBS). E.V.K. is supported through the intramural program of the U.S. National Institutes of Health.

## Author contributions

M.K. conceived the study. D.K. and M.K. performed sequence analyses. D.K., A.V., E.V.K., and M.K. interpreted the results. D.K. and M.K. wrote the first draft of the manuscript. All authors edited and approved the final version of the manuscript.

## Additional information

**Competing interests:** The authors declare no competing interests.

