## [Peer Review File · Nature Communications]

Reviewers' Comments:

Reviewer #1:

Remarks to the Author:

This manuscript presents a comprehensive comparative genomic / evolutionary analysis of Rep-encoding ssDNA (CRESS-DNA) virus sequences, including bacterial virus (phage) and plasmid sequences that encode homologs. Following an in-depth high-sensitivity search of public datasets for related sequences, the primary analysis is based on similarity clustering, followed by a more formal phylogenetic analysis (for some clusters), and a detailed analysis of motif / domain distributions. This latter analysis is used to support the contention that the clustering analysis has successfully captured key aspects of the evolutionary relationships among these clades.

The authors' primary claim is striking, exciting, and likely to be of broad interest. Namely that "the replication machinery of the CRESS-DNA viruses evolved, on multiple independent occasions, from the Rep proteins of bacterial [...] plasmids" by gaining an association with capsid proteins of eukaryotic +ssRNA viruses.

Overall this seems a well done and interesting piece of research that I enjoyed reading, but I have two broad reservations that may need consideration.

First, although the analysis seems compelling on the surface, the majority of conclusions are supported only by verbal arguments based on patterns in the data (e.g. clustering analysis) rather than formal statistical tests (such as likelihood ratio tests of alternative tree topologies) – although the authors do (in several places) make relevant caveats clear. On the one hand, the verbal arguments made are compelling and well presented, and the use of motif/domain acquisition almost as an informal 'rare derived character' argument lends a great deal of support. But on the other hand I think the manuscript lacks clear formal tests of each statement of (e.g.) the number of origins of each group – which I would want to see of such high-profile claims in a high-profile journal. Such formal tests may not be possible (divergence may be too high to give them power) and if that is the case, then this is the best analysis that can be done.

This leads me to my second point (one for the editor) – as the findings are interesting, and on balance perfectly reasonable, but perhaps not statistically 'watertight', is this more suitable for such a high profile broad-interest journal or for a top-end virus journal?

Other more minor comments are listed below:

- 1) Although generally well-written, some of the sentence are simple too long, convoluted, and lacking suitable punctuation. For example, the opening sentence of the abstract, and
- 2) It would be good to introduce basic details of the 'HUH' motif earlier on
- 3) P1 Line 4 – "Rapidly expanding" in terms of our knowledge/understanding – not the group itself
- 4) P4 Lines 20-22 and especially page 12 L18-19: CLANS is a big part of the work that is presented here, and it needs at least 3-4 sentences of explanation to give the reader some context for what is actually being done
- 5) P4 Line 32, P5L5; P9L22: "connect" – here and elsewhere words are used that might imply relationship when the analysis is only considering similarity. I think the authors should be more careful to be explicit about the type of 'connection' etc at each use.
- 6) P6L2 what is meant by 'largely consistent'?
- 7) ML Phylogenetic analysis – in general, were any attempts made to look for potential recombination within genes / domains?
- 8) P10L22-27. I found this to be confusing and/or poorly phrased. It's perfect reasonable that the 'filamentous bacteriophages' are considered as not a natural taxonomic group if replication genes don't support that relationships due to horizontal transfer (whereas structural genes do support the group). But then it makes no sense to say "by contrast, in the case of RNA viruses, it is the replication module which provides the most reliable scaffold for mapping evolutionary events" –

reliable in what sense? Why should we consider the gain and loss of structural proteins by a polymerase to be more reliable than the gain or loss of polymerases by a structural protein? When there are so few genes, surely the 'focal' one for taxonomy is arbitrary?

9) Page 11 Line 14-16 – has this claim regarding the origin of parvo/denso viruses been made before? If so, I think there should be a citation. If not, I'm surprised the authors do not make more of it here. It may be my taxonomic bias, but to me it seems very interesting!

10) P12L48-49 – I would like more on the branches that were removed. On what basis were they chosen, and what was the impact of their removal? – “We took out the branches that spoilt our story” would not be good, but “We took out branches that were inferred to be objectively 'rogue'” would be acceptable.

11) Figure 1 – please could a coloured key be added to the figure, rather than listing the colours in the legend text?

12) Figure 1 – perhaps the fonts and dots should be larger – there's a lot of white space

Reviewer #2:

Remarks to the Author:

The manuscript by Kazlauskas et al describe an analysis of the rapidly growing number of available rep proteins sequences found in bacteria, archaea, plasmids, and ssDNA viruses (prokaryotic and eukaryotic). The study presents a “grand unified theory” regarding the origin and relationship between Rep gene containing genetic elements. This evolution involves the late acquisition of the helicase domain situated downstream of the HUH endonuclease domain more typically found in prokaryotic elements followed by the capture of capsid sequences likely from the cDNA of an RNA virus.

These studies are now possible due to the massive influx of sequence data due to NGS and serve the important function of starting the assemble the big picture of viral evolution. Study reaches the not so surprising but now better supported conclusions that the very large group of CRESS-DNA viruses (which are dominant in many ecosystems) most likely originated as plasmids. The transition of plasmid to virus seems to have occurred on several occasions based on phylogenetic considerations.

The study was performed by experts in the field who thoroughly scanned databases to find all relevant sequences. Authors were also careful in their interpretations and when in doubt clear about possible alternative explanations.

This analysis of the diverse group of viruses containing a rep protein, will help the readers interested in viral evolution appreciate the lego-like recombination of protein domains from different sources over evolutionary time that resulted in today's immense level of viral diversity in ssDNA viruses.

Is it still correct to assume that ALL Rep with an helicase domain are from a eukaryotic host?

Reviewer #3:

Remarks to the Author:

Although this paper presents some very interesting ideas, and it is possible that the authors are right in their evolutionary scenarios, I have major concerns about the validity of the analysis undertaken and hence the robustness of the conclusions drawn. The situation is not helped because it is often difficult to tease apart 'fact' from 'idea' and 'conclusion' from 'hypothesis'. Hence, at present I cannot see conclusive evidence for the central claim of the paper that “the replication machinery of the CRESS-DNA viruses evolved, on multiple independent occasions, from the Rep proteins of bacterial rolling circle-replicating plasmids”. What we have are some tentative

evolutionary scenarios that need to be proven using far more robust data.

My single biggest concern about this paper is that the sequences analysed are so very divergent that it is in reality very, very hard to identify homology, let alone establish a robust evolutionary history. Identifying a few motifs, which are divergent in themselves, is NOT the same as establishing a reliable evolutionary history.

1. A lot is made of the CLANS analysis. However, this is really just a way to visualize Blast e-values and should not be used to infer evolutionary relationships. In particular, my reading of Fig 1 is different from the authors. What I see are no sequence similarity among some groups (i.e. *Rudiviridae* and IS200/IS605) and only very limited similarity between the parvoviruses and everything else and most importantly between the CRESS DNA viruses/Group 9 and the main cluster (supercluster 1?) of Rep proteins. The latter point is particularly important: I can only see one link from the group 9 sequences to the main group and one link from groups 1,2 and 3/CRESS DNA to the main group. If these groups were truly related - that is, they share common ancestry - we would expect to see far more connections. Hence, I see nothing in these data to rule out that the these residual similarities in protein sequence similarity are in fact due to convergent evolution or lateral gene transfer.

2. I have similar concerns about the phylogenetic analyses. What the authors fail to appreciate is that if you put sequences into a tree-building program you must get a tree back, even if there is no similarity/homology between them. It can be no other way. To my mind these sequences are simply so divergent in sequence that any phylogenetic analysis cannot be considered robust. Hence, from Figure 2 (A) while I can agree that there are some clear groups of REP sequences, they are so very divergent in sequence that their underlying phylogeny - the inter-group relationships - cannot be trusted and the bootstrap values are completely meaningless (and the same applies to Figure 5).

3. Most important of all, I have looked at the sequence alignments used to infer the trees in Figures 2 and 5 and these raise major concerns about the validity of all the analyses performed and hence the conclusions drawn. While it is certainly possible (although not proven) that the groups share some very short motifs, most of the alignment used to infer the trees is extremely non-robust and perhaps no better than by chance alone. To show this, I performed a GBlocks (analogous to TrimAL) pruning of highly divergent/dubious aligned regions in the alignments used to make the trees in Figures 2 and 5. In the case of Figure 2, whereas the TrimAL alignment used by the authors comprised 743 amino acid residues, GBlocks returned ZERO safely aligned sites under the default settings and only 59 residues if the most relaxed GBlocks settings were used. Obviously, 59 residues is too short for a meaningful analysis. The situation was even worse in the case of the Figure 5 alignment: whereas the TrimAL aligned used by the authors contained 508 amino acid results, GBlocks again returned zero safely aligned sites under the default settings and only 6 residues if the most relaxed GBlocks settings. In short, the phylogenetic trees and evolutionary history presented in this paper cannot be considered in any way robust. Again, motif and phylogeny are different things.

MINOR POINTS

1. Some of the text is very hard to follow. The authors move between viruses, clusters, super-clusters and groups and it is hard to tell what is what. Some consistent labelling on the figures would help.

2. The identification of 33 clusters in Figure 1 seems rather arbitrary to me, as does the identification of Supercluster 2.

3. I have no confidence in the tree rooting scheme presented in Figure 5.

RESPONSES TO THE REVIEWERS' COMMENTS

Reviewer #1 (Remarks to the Author):

This manuscript presents a comprehensive comparative genomic / evolutionary analysis of Rep-encoding ssDNA (CRESS-DNA) virus sequences, including bacterial virus (phage) and plasmid sequences that encode homologs. Following an in-depth high-sensitivity search of public datasets for related sequences, the primary analysis is based on similarity clustering, followed by a more formal phylogenetic analysis (for some clusters), and a detailed analysis of motif / domain distributions. This latter analysis is used to support the contention that the clustering analysis has successfully captured key aspects of the evolutionary relationships among these clades.

The authors' primary claim is striking, exciting, and likely to be of broad interest. Namely that "the replication machinery of the CRESS-DNA viruses evolved, on multiple independent occasions, from the Rep proteins of bacterial [...] plasmids" by gaining an association with capsid proteins of eukaryotic +ssRNA viruses.

Overall this seems a well done and interesting piece of research that I enjoyed reading, but I have two broad reservations that may need consideration.

First, although the analysis seems compelling on the surface, the majority of conclusions are supported only by verbal arguments based on patterns in the data (e.g. clustering analysis) rather than formal statistical tests (such as likelihood ratio tests of alternative tree topologies) – although the authors do (in several places) make relevant caveats clear. On the one hand, the verbal arguments made are compelling and well presented, and the use of motif/domain acquisition almost as an informal 'rare derived character' argument lends a great deal of support. But on the other hand I think the manuscript lacks clear formal tests of each statement of (e.g.) the number of origins of each group – which I would want to see of such high-profile claims in a high-profile journal. Such formal tests may not be possible (divergence may be too high to give them power) and if that is the case, then this is the best analysis that can be done.

RESPONSE: We certainly agree with the reviewer and appreciate the acknowledgement of the difficulties associated with the analyses of highly diverged sequences, as in the case of the datasets analyzed in this study. In the revised manuscript, to test the robustness of the PhyML tree, we performed the following additional analyses: (i) constructed maximum likelihood phylogenies using other tools, namely, RAxML and IQ-Tree, with alternative branch support methods, including the classical bootstrap and the more recently introduced ultrafast bootstrap procedures; (ii) reconstructed a maximum likelihood phylogeny using the 20-profile mixture model which, similar to Bayesian CAT models but in maximum likelihood framework, allows 20 substitution models along the sequences in the alignment; (iii) performed statistical analysis of the unconstrained and 3 constrained tree topologies using a number of statistical tests, including Approximately Unbiased (AU) test (PMID: 12079646), logL difference from the maximal logL in the set, RELL test (Kishino et al., J Mol Evol 1990, 31:151–160), one sided and weighted Kishino-Hasegawa (KH) tests (PMID: 2509717), Shimodaira-Hasegawa (SH) test (Shimodaira and Hasegawa, Mol Biol Evol, 1999, 8:1114-1116), weighted SH test, Expected Likelihood Weight (ELW) test (PMID: 11798428).

The IQ-Tree and RAxML trees had nearly identical topologies to that of the PhyML tree, although the branch support values estimated with the bootstrap procedure in RAxML tree were slightly lower than

the aBayes and ultrafast bootstrap values in the PhyML and IQ-Tree phylogenies, respectively. To account for potential differences in site-specific amino acid replacement patterns, we used the C20 mixture model. When C20 mixture model was used for phylogeny reconstruction, the obtained topology was nearly identical to that in the single-model maximum likelihood analyses (Figure 5 and Figure S5). To further scrutinize the robustness of the phylogenetic tree, we constructed a set of constrained trees with alternative topologies and compared these to the unconstrained tree using several statistical tests, including the approximately unbiased test (PMID: 12079646). All tests confidently rejected the trees with alternative topologies (Table S3). Collectively, these results indicate that the obtained tree topology is highly robust and is likely to accurately reflect the evolutionary history of Reprs encoded by CRESS-DNA viruses and plasmids.

Figure 5.

Supplementary figure 5

Table S3. Topology testing for the phylogenetic tree of Rep proteins.

Tree	AU ¹	deltaL ²	RELL ³	KH ⁴	SH ⁵	WKH ⁶	WSH ⁷	ELW ⁸
1 (Unc.)	0.997	0	0.995	0.996	1	0.996	1	0.995
2	0.00566	238.38	0.0036	0.00396	0.00661	0.00396	0.0113	0.00363
3	0.00315	262.38	0.00186	0.00231	0.00266	0.00231	0.00624	0.00185
4	5.77E-60	561.13	0	0	0	0	0	1.97E-36

Tree 1: Unconstrained topology;

Tree 2: plasmids and viruses form two monophyletic groups;

Tree 3: positions of the geminivirus/genomovirus clade and that including all other CRESS-DNA viruses are switched;

Tree 4: regrouped according to the host organisms, i.e., branch including plant-associated geminiviruses and genomoviruses is moved as a sister group to plant-associated nanoviruses/alphasatellites, animal-associated smacoviruses are grouped with circoviruses, whereas other unclassified groups of CRESS-DNA viruses are monophyletic.

¹ p-value of approximately unbiased (AU) test.

² logL difference from the maximal logL in the set.

³ bootstrap proportion using RELL method.

⁴ p-value of one sided Kishino-Hasegawa test.

⁵ p-value of Shimodaira-Hasegawa test.

⁶ p-value of weighted KH test.

⁷ p-value of weighted SH test.

⁸ Expected Likelihood Weight.

This leads me to my second point (one for the editor) – as the findings are interesting, and on balance perfectly reasonable, but perhaps not statistically ‘watertight’, is this more suitable for such a high profile broad-interest journal or for a top-end virus journal?

RESPONSE: As mentioned above, we performed a number of additional formal statistical analyses to support our conclusions. Given the available tools and datasets, the obtained phylogenies and the hypothesis, which they represent, are as robust as currently feasible, and in that sense, can be viewed as being 'watertight'.

Other more minor comments are listed below:

1) Although generally well-written, some of the sentence are simple too long, convoluted, and lacking suitable punctuation. For example, the opening sentence of the abstract, and

RESPONSE: Throughout the revised manuscript, we tried to split the overly long sentences.

2) It would be good to introduce basic details of the 'HUH' motif earlier on

RESPONSE: Done.

3) P1 Line 4 – "Rapidly expanding" in terms of our knowledge/understanding – not the group itself

RESPONSE: Rephrased.

4) P4 Lines 20-22 and especially page 12 L18-19: CLANS is a big part of the work that is presented here, and it needs at least 3-4 sentences of explanation to give the reader some context for what is actually being done

RESPONSE: We added the following explanation:

"CLANS is an implementation of the Fruchterman-Reingold force-directed layout algorithm, which treats protein sequences as point masses in a virtual multidimensional space, in which they attract or repel each other based on the strength of their pairwise similarities (CLANS p-values). Thus, evolutionarily more closely related sequences gravitate to the same parts of the map, forming distinct clusters.

5) P4 Line 32, P5L5; P9L22: "connect" – here and elsewhere words are used that might imply relationship when the analysis is only considering similarity. I think the authors should be more careful to be explicit about the type of 'connection' etc at each use.

RESPONSE: The text has been revised accordingly.

6) P6L2 what is meant by 'largely consistent'?

RESPONSE: The text has been rephrased.

7) ML Phylogenetic analysis – in general, were any attempts made to look for potential recombination within genes / domains?

RESPONSE: Indeed, the rep genes are known to be recombinogenic, with the regions encoding nuclease and helicase domains being exchanged by recombination between distantly related viruses sharing the same host. In our previous study (PMID: 29642587), we have specifically focused on this aspect and analyzed a large dataset of Reps encoded by cultivated and uncultivated CRESS DNA viruses. In the

current work, we considered only those Rep sequences in which both domains were found to co-evolve. This is mentioned in the manuscript (P 9, L 32-35)

8) P10L22-27. I found this to be confusing and/or poorly phrased. It's perfect reasonable that the 'filamentous bacteriophages' are considered as not a natural taxonomic group if replication genes don't support that relationships due to horizontal transfer (whereas structural genes do support the group). But then it makes no sense to say "by contrast, in the case of RNA viruses, it is the replication module which provides the most reliable scaffold for mapping evolutionary events" – reliable in what sense? Why should we consider the gain and loss of structural proteins by a polymerase to be more reliable than the gain or loss of polymerases by a structural protein? When there are so few genes, surely the 'focal' one for taxonomy is arbitrary?

RESPONSE: Thank you for pointing this out. Indeed, the original sentence was not well phrased and has been modified in the revised version. Which gene to choose for taxonomy is certainly arbitrary. What we meant to say, is that the RdRp in the case of RNA viruses is the most "convenient" marker, much like the rRNA gene for cellular organisms, for mapping evolutionary events. Plainly, the RdRp is the only gene that is universally conserved in all RNA viruses as we explicitly indicate in the revised manuscript.

9) Page 11 Line 14-16 – has this claim regarding the origin of parvo/denso viruses been made before? If so, I think there should be a citation. If not, I'm surprised the authors do not make more of it here. It may be my taxonomic bias, but to me it seems very interesting!

RESPONSE: Thank you for pointing this out. This is indeed a new observation, to our knowledge. In the revised manuscript, we added a couple of additional sentences on this topic: "Our findings further suggest that parvoviruses that have linear ssDNA genomes evolved directly from the CRESS-DNA viruses. Indeed, both Rep and capsid proteins of parvoviruses are homologous to those of CRESS-DNA viruses. Compared with the Reps involved in the rolling-circle replication, the Rep of parvoviruses lacks the joining activity that is used by the CRESS-DNA viruses to circularize progeny genomes. Instead, the parvovirus Rep remains covalently attached to the 5' ends of all viral DNA molecules (23). Conceivably, the loss of the diagnostic Motif 1 in the parvovirus Rep (20) resulted in the switch from the rolling-circle to the rolling-hairpin replication mode and led to substantial sequence diversification."

10) P12L48-49 – I would like more on the branches that were removed. On what basis were they chosen, and what was the impact of their removal? – "We took out the branches that spoilt our story" would not be good, but "We took out branches that were inferred to be objectively 'rogue'" would be acceptable.

RESPONSE: Only the clade including bacilladnaviruses was removed, because they are (i) highly divergent and form a long branch; (ii) there are only a small number of sequences in the clade, which likely contributed to the unstable position in phylogenies; (iii) based on motif analysis, their Reps appear to be recombinant. As a result, this group of viruses shows an unstable position in phylogenetic analyses and hence we removed it. For transparency, in the supplementary information, we show 2 trees including bacilladnaviruses from which the instability is obvious.

11) Figure 1 – please could a coloured key be added to the figure, rather than listing the colours in the legend text?

RESPONSE: Done.

12) Figure 1 – perhaps the fonts and dots should be larger – there’s a lot of white space

RESPONSE: We increased the fonts, but increasing the dots did not yield a satisfactory result.

Reviewer #2 (Remarks to the Author):

The manuscript by Kazlauskas et al describe an analysis of the rapidly growing number of available rep proteins sequences found in bacteria, archaea, plasmids, and ssDNA viruses (prokaryotic and eukaryotic). The study presents a “grand unified theory” regarding the origin and relationship between Rep gene containing genetic elements. This evolution involves the late acquisition of the helicase domain situated downstream of the HUH endonuclease domain more typically found in prokaryotic elements followed by the capture of capsid sequences likely from the cDNA of an RNA virus.

These studies are now possible due to the massive influx of sequence data due to NGS and serve the important function of starting the assemble the big picture of viral evolution. Study reaches the not so surprising but now better supported conclusions that the very large group of CRESS-DNA viruses (which are dominant in many ecosystems) most likely originated as plasmids. The transition of plasmid to virus seems to have occurred on several occasions based on phylogenetic considerations.

The study was performed by experts in the field who thoroughly scanned databases to find all relevant sequences. Authors were also careful in their interpretations and when in doubt clear about possible alternative explanations.

This analysis of the diverse group of viruses containing a rep protein, will help the readers interested in viral evolution appreciate the lego-like recombination of protein domains from different sources over evolutionary time that resulted in today’s immense level of viral diversity in ssDNA viruses.

RESPONSE: We appreciate the positive assessment of our work.

Is it still correct to assume that ALL Rep with an helicase domain are from a eukaryotic host?

RESPONSE: Indeed, our analysis shows that Reps with the superfamily 3 helicase domain have originated in bacteria and were only subsequently transferred to eukaryotic viruses.

Reviewer #3 (Remarks to the Author):

Although this paper presents some very interesting ideas, and it is possible that the authors are right in their evolutionary scenarios, I have major concerns about the validity of the analysis undertaken and hence the robustness of the conclusions drawn. The situation is not helped because it is often difficult to tease apart ‘fact’ from ‘idea’ and ‘conclusion’ from ‘hypothesis’. Hence, at present I cannot see conclusive evidence for the central claim of the paper that “the replication machinery of the CRESS-DNA viruses evolved, on multiple independent occasions, from the Rep proteins of bacterial rolling circle-

replicating plasmids". What we have are some tentative evolutionary scenarios that need to be proven using far more robust data.

RESPONSE: In the revised manuscript, we performed a number of additional analyses to support the robustness of the phylogenetic analysis presented in the original manuscript, as detailed below. The text has also been revised to make it more accessible.

We are not sure what would qualify as "far more robust data" to buttress our evolutionary reconstructions. Certainly, we present an evolutionary scenario rather than "facts" of actual evolutionary events. However, this scenario is built on a large, inclusive dataset of sequences from diverse taxa, and analyses were performed using state-of-the-art methods in comparative genomics and phylogenetics. It is highly unlikely that, in the near future, some data comes to light that would radically change the conclusions of this work, although specific refinements can be certainly expected. Regardless, we are convinced that the evolutionary scenario presented here will be useful to the community working on CRESS-DNA viruses, will provide a framework for further inquiries into the origin and evolution of this large class of viruses, and will be appreciated by a broad audience of researchers interested in virus evolution.

My single biggest concern about this paper is that the sequences analysed are so very divergent that it is in reality very, very hard to identify homology, let alone establish a robust evolutionary history. Identifying a few motifs, which are divergent in themselves, is NOT the same as establishing a reliable evolutionary history.

RESPONSE: Although it is indeed challenging to analyze divergent sequences, over the past decade, a number of computational tools for homology detection and phylogenetic reconstruction, including various evolutionary models accounting for heterogeneity across sites as well as a number of formal statistical tests, have been developed which now allow robust analyses of highly divergent sequences. These tools were used in this work.

1. A lot is made of the CLANS analysis. However, this is really just a way to visualize Blast e-values and should not be used to infer evolutionary relationships.

RESPONSE: Actually, CLANS is more than "just a way to visualize Blast e-values". Rather, it is a method that generates clusters of sequences that show closer relationship to each other than to other sequences in the analyzed set. We added the following explanation in the revised text: "CLANS is an implementation of the Fruchterman-Reingold force-directed layout algorithm, which treats protein sequences as point masses in a virtual multidimensional space, in which they attract or repel each other based on the strength of their pairwise similarities (CLANS p-values). Thus, evolutionarily more closely related sequences gravitate to same parts of the map, forming clusters of related sequences."

In particular, my reading of Fig 1 is different from the authors. What I see are no sequence similarity among some groups (i.e. Rudiviridae and IS200/IS605) and only very limited similarity between the parvoviruses and everything else and most importantly between the CRESS DNA viruses/Group 9 and the main cluster (supercluster 1?) of Rep proteins. The latter point is particularly important: I can only see one link from the group 9 sequences to the main group and one link from groups 1,2 and 3/CRESS DNA to the main group.

RESPONSE: This assessment appears to closely match the one which we made in the original text: the Rep diversity is split into 4 groupings: 2 orphan clusters, including Rudiviridae and IS200/IS605, respectively, and 2 superclusters. By far the largest, Supercluster 1 includes 24 clusters of Reps from diverse bacterial and archaeal plasmids, viruses and transposons, including eukaryotic Helitrons. By contrast, Supercluster 2 includes all the CRESS-DNA viruses and loosely connected parvoviruses as well as bacterial plasmids of groups 1 to 9 and pE194/pMV158-like plasmids, which are strongly connected to the CRESS-DNA viruses. We clearly state that there is either no or very few connections obvious at the sequence level between the 4 groupings (in particular, CRESS-DNA viruses/group 1-9 plasmids are not connected to supercluster 1). Hence, we do not find any significant discrepancy between our own account of the relationships between the compared genomes and that of the reviewer, and accordingly, we do not see this comment as pointing to any problem with our analysis.

If these groups were truly related - that is, they share common ancestry - we would expect to see far more connections. Hence, I see nothing in these data to rule out that these residual similarities in protein sequence similarity are in fact due to convergent evolution or lateral gene transfer.

RESPONSE: It is exactly our point that supercluster 1 and supercluster 2 are very distantly related. That is also why the rest of the manuscript focuses exclusively on supercluster 2 and, in particular, on the evolutionary relationships among bacterial plasmids of groups 1-9 and CRESS-DNA viruses. This said, the evolutionary relationships among the 4 groups is obvious from comparisons of the X-ray structures of the corresponding representative proteins (this is clearly stated in the text). Indeed, high-resolution structures of Rep and transposases are available for representatives of the 2 orphan clusters and multiple structures are available for each of the 2 superclusters. They all share the same overall structural fold, the same active site (including the 3 signature motifs) and the same catalytic mechanisms. Given these similarities, the possibility of convergence can be ruled out with confidence.

2. I have similar concerns about the phylogenetic analyses. What the authors fail to appreciate is that if you put sequences into a tree-building program you must get a tree back, even if there is no similarity/homology between them. It can be no other way. To my mind these sequences are simply so divergent in sequence that any phylogenetic analysis cannot be considered robust. Hence, from Figure 2 (A) while I can agree that there are some clear groups of REP sequences, they are so very divergent in sequence that their underlying phylogeny - the inter-group relationships - cannot be trusted and the bootstrap values are completely meaningless (and the same applies to Figure 5).

RESPONSE: It is surprising to read such a comment stating that state-of-the-art phylogenetic tools would produce well-supported trees from random sequences and that all the statistical tests “cannot be trusted and the bootstrap values are completely meaningless”. Effectively, by taking this position, one would dismiss the entire field of phylogenetic analysis as irrelevant. Respectfully, we think that this comment is unsubstantiated. The phylogenies in Figures 2 and 5 are based on unambiguous alignments of homologous sequences which were obtained by sequence similarity searches with statistically significant inclusion thresholds. All sequences and alignments were manually inspected for the presence and correct alignment of both the nuclease and helicase domains. Furthermore, we applied formal statistical tests to verify the robustness of the original, unconstrained tree and several constrained trees, in which we forced grouping of certain branches. In all cases, the constrained trees were confidently rejected. These new analyses are now included in the revised manuscript (see below and also the response to reviewer 1).

3. Most important of all, I have looked at the sequence alignments used to infer the trees in Figures 2

and 5 and these raise major concerns about the validity of all the analyses performed and hence the conclusions drawn. While it is certainly possible (although not proven) that the groups share some very short motifs, most of the alignment used to infer the trees is extremely non-robust and perhaps no better than by chance alone. To show this, I performed a GBLOCKS (analogous to TrimAL) pruning of highly divergent/dubious aligned regions in the alignments used to make the trees in Figures 2 and 5. In the case of Figure 2, whereas the TrimAL alignment used by the authors comprised 743 amino acid residues, GBLOCKS returned ZERO safely aligned sites under the default settings and only 59 residues if the most relaxed GBLOCKS settings were used. Obviously, 59 residues is too short for a meaningful analysis. The situation was even worse in the case of the Figure 5 alignment: whereas the TrimAL aligned used by the authors contained 508 amino acid results, GBLOCKS again returned zero safely aligned sites under the default settings and only 6 residues if the most relaxed GBLOCKS settings. In short, the phylogenetic trees and evolutionary history presented in this paper cannot be considered in any way robust. Again, motif and phylogeny are different things.

RESPONSE: Although, historically, GBLOCKS was one of the first alignment trimming tools to be developed (back in 2000), it has important shortcomings compared to the Trimal method that was used in our study. GBLOCKS was designed for moderately divergent sequences and, unlike Trimal, it does not use a substitution matrix or model of evolution and does not adapt parameters for particular data sets. In the case of divergent sequences, GBLOCKS largely removes the columns with gaps and its effective use depends on the careful setting of several parameters. Importantly, it has been demonstrated using various real and simulated datasets that the more one trims the alignment the worse the tree gets (PMID: 26031838). In fact, it has been shown that GBLOCKS trimming produces the worst trees compared to other similar tools (PMID: 26031838). The greedy trimming tools, such as GBLOCKS, could have been useful and justifiable in the early days of phylogenetic analyses, when powerful phylogenetic reconstruction methods and adequate substitution matrices were not yet available. However, at present, the powerful state-of-the-art methods for trimming multiple sequence alignments and phylogenetic analysis allow reliable reconstruction of the evolutionary history of even highly divergent sequences.

Furthermore, in the revised manuscript, we include a number of additional analyses to test the robustness of the phylogenetic trees, which is central to our evolutionary scenario. In particular, we performed the following additional analyses: (i) constructed maximum likelihood phylogenies using other tools, namely, RAxML and IQ-Tree, with alternative branch support methods, including the classical bootstrap and the more recently introduced ultrafast bootstrap procedures; (ii) reconstructed a maximum likelihood phylogeny using the 20-profile mixture model which, similar to Bayesian CAT models but in maximum likelihood framework, allows 20 substitution models along the sequences in the alignment; (iii) performed statistical analysis of the unconstrained and 3 constrained tree topologies using a number of statistical tests, including Approximately Unbiased (AU) test (PMID: 12079646), logL difference from the maximal logL in the set, RELL test (Kishino et al., J Mol Evol 1990, 31:151–160), one sided and weighted Kishino-Hasegawa (KH) tests (PMID: 2509717), Shimodaira-Hasegawa (SH) test (Shimodaira and Hasegawa, Mol Biol Evol, 1999, 8:1114-1116), weighted SH test, Expected Likelihood Weight (ELW) test (PMID: 11798428).

The IQ-Tree and RAxML trees had nearly identical topology to that of the PhyML tree although the branch support values estimated with the bootstrap procedure in RAxML tree were slightly lower than the aBayes and ultrafast bootstrap values in the PhyML and IQ-Tree trees, respectively. To account for potential differences in site-specific amino acid replacement patterns, we used the C20 mixture model. When the C20 mixture model was used for phylogeny reconstruction, the obtained topology was nearly

identical to that in the single-model maximum likelihood analyses (Figure 5 and Figure S5). To further scrutinize the robustness of the phylogenetic tree, we constructed a set of constrained trees with alternative topologies and compared them to the unconstrained tree using several statistical tests, including the approximately unbiased test (PMID: 12079646). All tests rejected the trees with alternative topologies (Table S3). Collectively, these results indicate that the obtained tree topology is highly robust and is likely to accurately reflect the evolutionary history of Repls encoded by CRESS-DNA viruses and plasmids.

MINOR POINTS

1. Some of the text is very hard to follow. The authors move between viruses, clusters, super-clusters and groups and it is hard to tell what is what. Some consistent labelling on the figures would help.

RESPONSE: Whenever possible, we tried to minimize the mention of groups. However, the categories listed by the reviewer are not synonymous and, unfortunately, cannot be avoided. We made an additional effort to revise the manuscript for clarity.

2. The identification of 33 clusters in Figure 1 seems rather arbitrary to me, as does the identification of Supercluster 2.

RESPONSE: We would like to assure the reviewer that there was nothing arbitrary about the identification of the 33 clusters. As is stated in the manuscript "clusters were identified using a convex clustering algorithm (p-value threshold of 1e-08) implemented in CLANS". The composition of each cluster is indicated in the supplementary table.

3. I have no confidence in the tree rooting scheme presented in Figure 5.

RESPONSE: The placement of the root is discussed at length in the manuscript: "Analysis of the SF3 helicase domains suggests that Repls of pE194/pMV158-like plasmids are ancestral rather than derived forms. The alternative possibility, namely, that Repls of pE194/pMV158-like plasmids have lost the helicase domain, cannot be ruled out at the moment. However, the fact that the helicase domain has not been lost in any of the numerous known groups of CRESS-DNA viruses or in plasmid groups pCRESS1 to pCRESS9, suggests that, once acquired, the helicase domain becomes integral for efficient plasmid/viral genome replication. Thus, the direct connection between the pE194/pMV158-like Repls and those of the 'YLxH' supergroup (Figure 1) implies that the former group is an adequate outgroup for the phylogeny of Repls from bacterial plasmids and CRESS-DNA viruses."

Reviewers' Comments:

Reviewer #2:

Remarks to the Author:

The revised manuscript by Kazlauskas et al is an important "big picture" analysis of the evolution of a large fraction of ssDNA viruses and related bacterial plasmids. This is a dense but very nicely performed and written study that clarifies the likely origin of not only eukaryotic CRESS-DNA viruses and parvoviruses but also dsDNA genomes of polyomaviruses and papillomaviruses from prokaryotic mobile genetic elements. This study will also be a great help in generating a unified theory of viral evolution and classifying together what are now unconnected viral families.

Authors should consider the possibility that the Gastropod associated GasCSV-like viruses are actually bacterial elements rather than eukaryotic tropic which would provide a more parsimonious explanation for their descent from a pCRESS2.

Maybe spell out a bit more forcefully that Reps of bacterial plasmids can include the SF3 helicase domain but that bacterial viruses never do (if correct).

Despite the discussed finding of smacovirus CRISPR sequences in archaea is it the current conclusion of the authors that all CRESS-DNA infect eukaryotes?

P4 line 9. Would it be wrong to insert: ...connections between viruses with small DNA Rep expressing genomes and capsid-less mobile genetic elements?

Can author venture an opinion on the origin of anelloviruses? Maybe another paper looking at anellovirus related plasmid?

Could also discuss that the absence of the SF3 helices domain can not be used as evidence that a Rep encoding circular genome must be of eukaryotic origin since it is now clear that bacterial plasmid Reps have SF3 helicases. Can authors suggest a way to differentiate bacterial from eukaryotic Rep beside phylogenetic affinity to the clades in figure 5? The alignment used to make figure 5 would be useful to many others if available as supplemental data so that future sequences can be readily compared to these clades.

Eric Delwart

Reviewer #4:

Remarks to the Author:

Overall comments:

Combining phylogenetic trees with protein clustering networks is a good way to study the evolutionary history of a highly divergent group of proteins, such as the Rep group. The statistical support for the tree topology and the similarity to patterns in the SSN provide convincing evidence that the phylogenetic trees are likely good models for this past evolution. The constrained tree topologies made for good comparisons between different evolutionary scenarios. However, the analysis is hard to follow. It is sometimes unclear which dataset the authors are currently examining, and it is hard for the reader to keep track of all the different groups being analyzed. Changing the names of these groups also means it is difficult for the reader to trace a group the entire way through the paper. The creation of a summary table and a more detailed methods section would help with these issues.

Specific comments:

p3 line12-17: This sentence contains too many thoughts, needs to be broken up.

p4 line16-19: Why were the Mob relaxases excluded? I assume they do not contain an HUH endonuclease? The authors should briefly elaborate on this point in the text.

p4 line20-22: UniRef is well-known, so including the name of the database would be good here.

P4 In 21: What is a jackhammer iteration? This is undefined jargon with no citation.

p4 line32-33: Are there any citations to support the idea that similar folds (i.e., protein structures as mentioned here) cannot result from convergent evolution?

p5 line13-15: I cannot tell from this sentence if the IS91 transposons and Helitrons are mixed within these two clusters or form distinct clusters.

p6 line5: The "only" in this line suggests that the authors were surprised that so few of the remaining Reps were from prophages. Is this the case, or were the authors simply reporting the numbers?

p6 line43-44: The methods mention a second set of bacterial Reps obtained from nr90. I believe this is when the authors begin to utilize this second dataset, but it is never mentioned in the results.

p6 line40-41: What is the domain organization like in prokaryotic plasmids and other viruses? A reference here would be useful.

p6 line51: Reps of groups 4-8 cannot be readily identified in the figure without labels.

p7 lines 1 and 28: Both of these direct the reader to figures that contain names that have not yet been introduced.

p7 line16: How were these Reps selected?

p7 line22-25: What constitutes an element?

p7 lines44-45: The protein SSN has already demonstrated the relatedness between the Reps of from the pCRESS groups to CRESS-DNA viruses. It might be better to use the pCRESS names from the beginning to minimize the number of names and name changes.

p7 line48: All of the motif analysis should be in the same section. The earlier paragraph (p7 lines1-10) should be moved to this section.

p9 line28-29: I am unsure that this claim is true. The helicase domain does not have to be integral to be kept, just useful.

p9 line37: There is an issue with the reference formatting.

p9 line32-37: It is unclear from this description which groups are included in the phylogeny. Is it all of the sequences from SC2 combined with reference sequences from citation 62? Or is it the bacterial Reps from nr90 combined with the reference sequences?

p10 line1: I am unclear on why the sharing of gene content between pCRESS7 and pCRESS9 is related to pCRESS9 Reps evolving from genomo- and geminiviridae Reps. Furthermore, what gene content do they share? The authors should also direct the readers to a specific tree or trees in figure S3.

p11 line17: I believe this is an additional result, not a converse one.

p11 line30-31: Are these viral families thought to be more ancient? Or is this speculation based on the phylogeny?

p11 line42: citation?

p12 line 32: The results from this paper are strong, and the authors' analyses have certainly shed additional light on the evolution of CRESS-DNA viruses. However, the authors themselves have acknowledged that there are aspects of their phylogeny that contradict their clustering networks. Additionally, the conclusion that CRESS-DNA viruses emerged three separate times is reliant not only on the Rep phylogeny, but also on the capsid genes. Based on the phylogeny, there is also a scenario in which the common ancestor of clades 1 and 2 is an extinct viral lineage, meaning that CRESS-DNA viruses would have emerged twice, rather than three times. The addition of capsid proteins to the evolutionary story is certainly useful and does imply three separate emergences, I would not say that the results are definitive.

p13 line11: How were these representative Repls chosen?

p13 line13: How few is few?

p13 line27: There is only one dataset being discussed, did the authors mean to use 'resulting' instead of 'latter'?

p13 line35: What are the latter profiles?

p13 line39: Were the phylogenies constructed from the entire Rep gene, or only with the HUH endonuclease domain?

p13 line48: Improperly formatted citation.

p14 line3: Should this refer to supplementary figure 6, rather than 4?

p14 line6: Improperly formatted citation.

p14 line6-9: It is unclear which trees were being tested. Was each additional tree tested against the original PhyML tree?

p14 line12: Improperly formatted citation.

p15 line9: Are the dots in figure 2B colored based on the tree clades or based on a clustering p-value or other metric?

p15 line17: The figure caption should mention why some of the CRESS-DNA virus clades now have group names/numbers.

p15 line5: The authors constructed many trees with many programs, so the software used to produce the tree in figure 5 should be mentioned in the caption for clarity.

Figure Comments:

Figure 3: This figure needs better labeling. The source gene/region of the motifs should be labeled at the top of the figure so that the reader does not have to search the main text for which motifs belong to which gene. Is there an order to the Rep groups in this figure, i.e. are more similar groups shown next to each other? The pCRESS groups have not yet been introduced at the time that the reader is directed to the figure. Thus, the reader is not yet able to read the figure. This figure may also be better in the supplement. Similar motifs are important for inferring evolutionary relatedness, but the reader cannot see which motifs are more related. Percent identity shared between groups may be a good metric to include in the main text, because it is easily understood

by readers. Sequence logos for motifs examined in the results could still be included in the main text.

Figure 4: It may be useful to color the proteins by source, as in figure 1.

Figure 5: This figure needs a color legend, in addition to or instead of the text in the caption. It would also be useful for the reader if the clades (1 and 2) were clearly labeled on the figure.

Figure 6: If possible, it may be useful to color bacterial and viral groups differently.

Reviewer #5:

Remarks to the Author:

In the present article "Natural history of the rolling-circle replicons: Multiple origins of prokaryotic and eukaryotic ssDNA viruses from bacterial plasmids" by Krupovic et al., the authors performed a comprehensive evolutionary analysis of the Rep protein sequences found in microbes, plasmids, and ssDNA viruses. The authors showed a clear exchange of those Rep genes between viruses and plasmids and hypothesized about their origin and evolution using the state-of-the-art tools in phylogeny and protein evolution.

The flow of the paper is very well structured and the main result is very exciting demonstrating a bidirectional transition between some types of plasmids and viruses. This finding is very interesting and important for example to researchers in the metagenomics field since there are numerous "ambiguous" partial sequences very hard to classify as one or another.

The authors have been able to show computational evidence to believe that CRESS-DNA viruses evolved from plasmids --after a capsid acquisition event, Parvoviruses evolved from CRESS-DNA viruses, some plasmids derive from CRESS-DNA viruses, etc.

In my opinion, the results are solid and well supported. Both, the response to the reviewers (which definitely have positively impacted the quality of the article) and the paper are very well stated and present cutting-edge results.

I would just ask for a more detailed method/supplemental sections where all the sequences (not only the accession numbers) and all the commands used in the publicly available software (e.g. MAFFT, CLANS, CD-HIT, IQ-Tree ...) were explicitly stated for a perfect reproducibility. Also, in this transparency exercise, the created trees could be shared (as an iTOL shared link).

POINT-BY-POINT RESPONSES TO THE REVIEWERS' COMMENTS

Reviewer #2 (Remarks to the Author):

The revised manuscript by Kazlauskas et al is an important "big picture" analysis of the evolution of a large fraction of ssDNA viruses and related bacterial plasmids. This is a dense but very nicely performed and written study that clarifies the likely origin of not only eukaryotic CRESS-DNA viruses and parvoviruses but also dsDNA genomes of polyomaviruses and papillomaviruses from prokaryotic mobile genetic elements. This study will also be a great help in generating a unified theory of viral evolution and classifying together what are now unconnected viral families.

RESPONSE: We are very grateful to Prof. Delwart for his comments which have helped to further improve and clarify our manuscript.

Authors should consider the possibility that the Gastropod associated GasCSV-like viruses are actually bacterial elements rather than eukaryotic tropic which would provide a more parsimonious explanation for their descent from a pCRESS2.

RESPONSE: We added the following sentence in the Discussion:
"Given the relatively close relationship between pCRESS2 and GasCSV-like viruses, it appears possible that viruses of the latter group infect bacteria rather than eukaryotes."

Maybe spell out a bit more forcefully that Repls of bacterial plasmids can include the SF3 helicase domain but that bacterial viruses never do (if correct).

RESPONSE: In the 3rd paragraph of the revised Introduction we explicitly state:
"By contrast, none of the bacterial or archaeal ssDNA viruses isolated to date encodes a Rep fused to a helicase domain".

Despite the discussed finding of smacovirus CRISPR sequences in archaea is it the current conclusion of the authors that all CRESS-DNA infect eukaryotes?

RESPONSE: We cannot be certain of this. For instance, GasCSV-like viruses might be associated with bacteria (see the response above). We also added the following sentence in the Discussion:
"We note, however, that hosts for CRESSVI-6 are currently unknown and might include organisms from any of the 3 cellular domains of life."

P4 line 9. Would it be wrong to insert: ...connections between viruses with small DNA Rep expressing genomes and capsid-less mobile genetic elements?

RESPONSE: Thank you for the suggestion. We have specified this in the following way:
"Similarly, the origins of bacterial and archaeal ssDNA viruses **replicating by the rolling-circle mechanism** can be traced to different families of prokaryotic plasmids, emphasizing tight evolutionary connections between viruses and capsid-less mobile genetic elements (MGE)."

Can author venture an opinion on the origin of anelloviruses? Maybe another paper looking at anellovirus related plasmid?

RESPONSE: Unfortunately, at this point, we cannot offer any scenario for the origin of anelloviruses. Certainly, this is a very interesting subject for further investigation.

Could also discuss that the absence of the SF3 helices domain can not be used as evidence that a Rep encoding circular genome must be of eukaryotic origin since it is now clear that bacterial plasmid Reps have SF3 helicases. Can authors suggest a way to differentiate bacterial from eukaryotic Rep beside phylogenetic affinity to the clades in figure 5? The alignment used to make figure 5 would be useful to many others if available as supplemental data so that future sequences can be readily compared to these clades.

RESPONSE: This is now mentioned:

“Furthermore, given that the SF3 helicase domain is now found in Reps of diverse bacterial replicons, this signature should be considered with caution when attributing the viral genomes discovered by metagenomics to particular hosts.”

As suggested, alignments used to generate figures 2 and 5 are now provided in the Supplementary data file 2.

Reviewer #4 (Remarks to the Author):

Overall comments:

Combining phylogenetic trees with protein clustering networks is a good way to study the evolutionary history of a highly divergent group of proteins, such as the Rep group. The statistical support for the tree topology and the similarity to patterns in the SSN provide convincing evidence that the phylogenetic trees are likely good models for this past evolution. The constrained tree topologies made for good comparisons between different evolutionary scenarios. However, the analysis is hard to follow. It is sometimes unclear which dataset the authors are currently examining, and it is hard for the reader to keep track of all the different groups being analyzed. Changing the names of these groups also means it is difficult for the reader to trace a group the entire way through the paper. The creation of a summary table and a more detailed methods section would help with these issues.

RESPONSE: We are very grateful to this reviewer for all the constructive comments. In the revised manuscript we have minimized the switching between names by eliminating the references to “groups” and replacing them with the “pCRESS” nomenclature. We have also revised the manuscript for clarity and provided additional details in the methods section.

Specific comments:

p3 line12-17: This sentence contains too many thoughts, needs to be broken up.

RESPONSE: The sentence has been split into two.

p4 line16-19: Why were the Mob relaxases excluded? I assume they do not contain an HUH endonuclease? The authors should briefly elaborate on this point in the text.

RESPONSE: Mob relaxases have a circular permutation, so that Motif 3 of the nuclease domain is found upstream of the Motif 1. This complicates sequence comparisons. We added the following explanation: “Enzymes in this family encompass circularly permuted conserved motifs which complicates their sequence-based comparison with the HUH endonucleases involved in DNA replication or transposition (Refs 19,22).”

p4 line20-22: UniRef is well-known, so including the name of the database would be good here.

RESPONSE: UniRef database is now explicitly mentioned.

P4 ln 21: What is a jackhammer iteration? This is undefined jargon with no citation.

RESPONSE: Jackhammer is a PSI-BLAST analog from the HMMER software suite. This is now clarified in the text.

p4 line32-33: Are there any citations to support the idea that similar folds (i.e., protein structures as mentioned here) cannot result from convergent evolution?

RESPONSE: We added the following explanation "...because protein structures are typically more conserved than the corresponding sequences" and added a reference to "Grishin NV. Fold change in evolution of protein structures. J Struct Biol. 2001;134(2-3):167-85."

p5 line13-15: I cannot tell from this sentence if the IS91 transposons and Helitrons are mixed within these two clusters or form distinct clusters.

RESPONSE: The clusters including IS91 and Helitrons are disconnected. We revised the sentence for clarity.

p6 line5: The "only" in this line suggests that the authors were surprised that so few of the remaining Repts were from prophages. Is this the case, or were the authors simply reporting the numbers?

RESPONSE: This was indeed somewhat surprising. Given that the only extrachromosomal elements in this cluster were phages, we expected that most of the 'bacterial' sequences will be prophages. However, only 10% of these were prophages.

p6 line43-44: The methods mention a second set of bacterial Repts obtained from nr90. I believe this is when the authors begin to utilize this second dataset, but it is never mentioned in the results.

RESPONSE: Indeed. This is now clarified in the main text: "Thus, we used representative members of SC2 to search for homologs in the nr90 database (see Methods)."

p6 line40-41: What is the domain organization like in prokaryotic plasmids and other viruses? A reference here would be useful.

RESPONSE: We added the following explanation: "..., which typically do not contain enzymatic domains other than the HUH endonuclease (Ref 4)."

p6 line51: Repts of groups 4-8 cannot be readily identified in the figure without labels.

RESPONSE: The 'YLxH supergroup' is now labeled as 'YLxH supergroup (pCRESS4-8)'.

p7 lines 1 and 28: Both of these direct the reader to figures that contain names that have not yet been introduced;

p7 lines44-45: The protein SSN has already demonstrated the relatedness between the Repts of from the pCRESS groups to CRESS-DNA viruses. It might be better to use the pCRESS names from the beginning to minimize the number of names and name changes.

RESPONSE: Thank you for pointing this out. To address both points, we have introduced the term 'pCRESS' for virus-like plasmid Repts earlier in the section and eliminated all references to 'groups'.

p7 line16: How were these Repls selected?

RESPONSE: When Repls were from wgs contigs, rather than closed genomes, the contigs were inspected manually and only those in which the rep gene was located 10 or more kb away from the contig terminus were retained.

p7 line22-25: What constitutes an element?

RESPONSE: The “element” corresponds to “integrated mobile genetic element” (this is now clarified in the text). Generally, these are viruses or plasmids which have been integrated into the host genome via homologous recombination between the attP site carried by the element DNA and the attB site on the bacterial chromosome. Following the recombination, the integrated element is flanked by chimeric attL and attR attachment sites, which appear as direct repeats.

p7 line48: All of the motif analysis should be in the same section. The earlier paragraph (p7 lines1-10) should be moved to this section.

RESPONSE: This paragraph was moved, as suggested.

p9 line28-29: I am unsure that this claim is true. The helicase domain does not have to be integral to be kept, just useful.

RESPONSE: We changed ‘integral’ in “helicase domain becomes integral for efficient plasmid/viral genome replication” to ‘important’.

p9 line37: There is an issue with the reference formatting.

RESPONSE: Actually, the citation is properly formatted. When citation follows a number, it has to be preceded by ‘Ref’.

p9 line32-37: It is unclear from this description which groups are included in the phylogeny. Is it all of the sequences from SC2 combined with reference sequences from citation 62? Or is it the bacterial Repls from nr90 combined with the reference sequences?

RESPONSE: The sentence has been revised:

“For phylogenetic analyses, we used a dataset of SC2 Repls, excluding Repls of Parvoviridae and CRESS-DNA viruses which were previously judged to be chimeric with respect to their nuclease and helicase domains (ref 64)...”

p10 line1: I am unclear on why the sharing of gene content between pCRESS7 and pCRESS9 is related to pCRESS9 Repls evolving from genomovirus- and geminiviridae Repls. Furthermore, what gene content do they share? The authors should also direct the readers to a specific tree or trees in figure S3.

RESPONSE: pCRESS7 and pCRESS9 from mollicutes share 3 proteins not found in elements from other families, namely, the copy number control protein, conserved hypothetical protein and PRK06752-like SSB protein. This information has been added to the text, and a pointer to Supplementary figure 3g and 3i is now included.

p11 line17: I believe this is an additional result, not a converse one.

RESPONSE: “Conversely” has been deleted.

p11 line30-31: Are these viral families thought to be more ancient? Or is this speculation based on the phylogeny?

RESPONSE: This inference is based on the fact that viruses in this assemblage infect hosts from different eukaryotic kingdoms. Namely, nanoviruses infect plants, whereas circoviruses infect animals.

p11 line42: citation?

RESPONSE: A reference to a recent review on CRESS-DNA viruses has been added (PMID: 30635078).

p12 line 32: The results from this paper are strong, and the authors’ analyses have certainly shed additional light on the evolution of CRESS-DNA viruses. However, the authors themselves have acknowledged that there are aspects of their phylogeny that contradict their clustering networks. Additionally, the conclusion that CRESS-DNA viruses emerged three separate times is reliant not only on the Rep phylogeny, but also on the capsid genes. Based on the phylogeny, there is also a scenario in which the common ancestor of clades 1 and 2 is an extinct viral lineage, meaning that CRESS-DNA viruses would have emerged twice, rather than three times. The addition of capsid proteins to the evolutionary story is certainly useful and does imply three separate emergences, I would not say that the results are definitive.

RESPONSE: We agree with the reviewer. Much remains to be done to fully understand the origins and evolution of CRESS-DNA viruses. We have removed “definitive”.

p13 line11: How were these representative Reps chosen?

RESPONSE: Representative Reps were selected as queries for homology searches based on exhaustive review of literature on the HUH superfamily (e.g., refs 19,20,38,64). This is now mentioned in the Methods section.

p13 line13: How few is few?

RESPONSE: Groups with less than 10 homologs. Now clarified in the text.

p13 line27: There is only one dataset being discussed, did the authors mean to use ‘resulting’ instead of ‘latter’?

RESPONSE: Corrected.

p13 line35: What are the latter profiles?

RESPONSE: Changed to “the resulting”.

p13 line39: Were the phylogenies constructed from the entire Rep gene, or only with the HUH endonuclease domain?

RESPONSE: Now specified that “The resulting alignments covered both HUH and SF3 (where available) domains”.

p13 line48: Improperly formatted citation.

RESPONSE: Actually, the citation is properly formatted. When citation follows a number, it has to be preceded by 'Ref'.

p14 line3: Should this refer to supplementary figure 6, rather than 4?

RESPONSE: Indeed. Corrected.

p14 line6: Improperly formatted citation.

RESPONSE: Actually, the citation is properly formatted. When citation follows a number, it has to be preceded by 'Ref'.

p14 line6-9: It is unclear which trees were being tested. Was each additional tree tested against the original PhyML tree?

RESPONSE: Yes, each constrained tree was tested against the original unconstrained PhyML tree. Now clarified in the text:

“As an unconstrained tree, we used the original PhyML tree (Figure 5), which was tested against each of the constrained trees”.

p14 line12: Improperly formatted citation.

RESPONSE: Corrected.

p15 line9: Are the dots in figure 2B colored based on the tree clades or based on a clustering p-value or other metric?

RESPONSE: The nodes belonging to the same cluster are colored with the same color and correspond to the clades shown in panel A. This sentence has been added to the figure legend and the clades in panel A have now been colored to match the colors in panel B.

p15 line17: The figure caption should mention why some of the CRESS-DNA virus clades now have group names/numbers.

RESPONSE: We added the following explanation in the legend:

“Groups of unclassified CRESS-DNA viruses are referred to as CRESSV1 through CRESSV6 (Ref 64).”

p15 line5: The authors constructed many trees with many programs, so the software used to produce the tree in figure 5 should be mentioned in the caption for clarity.

RESPONSE: Now mentioned.

Figure Comments:

Figure 3: This figure needs better labeling. The source gene/region of the motifs should be labeled at the top of the figure so that the reader does not have to search the main text for which motifs belong to which gene. Is there an order to the Rep groups in this figure, i.e. are more similar groups shown next to each other? The pCRESS groups have not yet been introduced at the time that the reader is directed to the figure. Thus, the reader is not yet able to read the figure. This figure may also be better in the supplement. Similar motifs are important for inferring evolutionary relatedness, but the reader cannot see which motifs

are more related. Percent identity shared between groups may be a good metric to include in the main text, because it is easily understood by readers. Sequence logos for motifs examined in the results could still be included in the main text.

RESPONSE: As suggested, we delineated the HUH endonuclease and the SF3 helicase domains at the top of the figure. We also clarified in the legend that the Rep groups are order according to the pairwise similarity in the aligned motifs, starting with the pE194/pMV158-like plasmids. As mentioned above, the pCRESS groups are now introduced early in the Results section and all references to 'groups' have been eliminated. However, we prefer retaining the figure as a main display item, because it is extensively discussed in the text.

Figure 4: It may be useful to color the proteins by source, as in figure 1.

RESPONSE: The nodes are now colored by source as in Figure 1.

Figure 5: This figure needs a color legend, in addition to or instead of the text in the caption. It would also be useful for the reader if the clades (1 and 2) were clearly labeled on the figure.

RESPONSE: The color key has been added to the figure (and removed from the caption) and the two clades are now also indicated, as suggested.

Figure 6: If possible, it may be useful to color bacterial and viral groups differently.

RESPONSE: We considered this option and concluded that additional colors make the figure confusing.

Reviewer #5 (Remarks to the Author):

In the present article "Natural history of the rolling-circle replicons: Multiple origins of prokaryotic and eukaryotic ssDNA viruses from bacterial plasmids" by Krupovic et al., the authors performed a comprehensive evolutionary analysis of the Rep protein sequences found in microbes, plasmids, and ssDNA viruses. The authors showed a clear exchange of those Rep genes between viruses and plasmids and hypothesized about their origin and evolution using the state-of-the-art tools in phylogeny and protein evolution.

The flow of the paper is very well structured and the main result is very exciting demonstrating a bidirectional transition between some types of plasmids and viruses. This finding is very interesting and important for example to researchers in the metagenomics field since there are numerous "ambiguous" partial sequences very hard to classify as one or another.

The authors have been able to show computational evidence to believe that CRESS-DNA viruses evolved from plasmids --after a capsid acquisition event, Parvoviruses evolved from CRESS-DNA viruses, some plasmids derive from CRESS-DNA viruses, etc.

In my opinion, the results are solid and well supported. Both, the response to the reviewers (which definitely have positively impacted the quality of the article) and the paper are very well stated and present cutting-edge results.

RESPONSE: We thank the reviewer for positive assessment of our work.

I would just ask for a more detailed method/supplemental sections where all the sequences (not only the accession numbers) and all the commands used in the publicly available software (e.g. MAFFT, CLANS, CD-HIT, IQ-Tree ...) were explicitly stated for a perfect reproducibility. Also, in this transparency exercise, the created trees could be shared (as an iTOL shared link).

RESPONSE: The methods section has been revised and all the parameters used for various software packages are now indicated. In the revised manuscript, we included the alignments (in fasta format) used to generate trees shown in Figures 2 and 5 as well as the trees shown in Figures 2, 5, S5a-c and S6a-b (in Newick format) in the Supplementary data file 2.

Reviewers' Comments:

Reviewer #5:

Remarks to the Author:

The updated version of the paper as well as the response to referees significantly improved the manuscript. In my opinion, it is suitable for publication.